# The genetic relationships between brain structure and schizophrenia

Eva-Maria Stauffer [1] ✉, Richard A. I. Bethlehem [1,2], Lena Dorfschmidt [1], Hyejung Won [3], Varun Warrier [1,2,5] & Edward T. Bullmore[1,4,5]

Genetic risks for schizophrenia are theoretically mediated by genetic effects on brain structure but it has been unclear which genes are associated with both schizophrenia and cortical phenotypes. We accessed genome-wide association studies (GWAS) of schizophrenia ($N = 69,369$ cases; 236,642 controls), and of three magnetic resonance imaging (MRI) metrics (surface area, cortical thickness, neurite density index) measured at 180 cortical areas ($N = 36,843$, UK Biobank). Using Hi-C-coupled MAGMA, 61 genes were significantly associated with both schizophrenia and one or more MRI metrics. Whole genome analysis with partial least squares demonstrated significant genetic covariation between schizophrenia and area or thickness of most cortical regions. Genetic similarity between cortical areas was strongly coupled to their phenotypic covariance, and genetic covariation between schizophrenia and brain phenotypes was strongest in the hubs of structural covariance networks. Pleiotropically associated genes were enriched for neurodevelopmental processes and positionally concentrated in chromosomes 3p21, 17q21 and 11p11. Mendelian randomization analysis indicated that genetically determined variation in a posterior cingulate cortical area could be causal for schizophrenia. Parallel analyses of GWAS on bipolar disorder, Alzheimer's disease and height showed that pleiotropic association with MRI metrics was stronger for schizophrenia compared to other disorders.

Recent genome-wide association studies (GWAS) have confirmed that human brain structure is heritable[1,2] and the number of genetic loci associated with variation in brain phenotypes, typically measured by magnetic resonance imaging (MRI), has increased as the scale of cohorts with both genetic and MRI data available has become larger[1,2]. Growing insight into the genetic architecture of the human brain raises the question of whether brain structural variation in the population is associated with genes that are also significantly associated with schizophrenia and other neuropsychiatric disorders[3–6]. It is a central expectation of many biological theories of schizophrenia that genetic effects on schizophrenia are mechanistically or proximally mediated by the effects of the same genes on brain structure and function[7]; but to date, there has been limited direct evidence for this prediction[3–6,8].

Here, we aimed to identify genes that are associated with both brain MRI phenotypes and schizophrenia. We reasoned that identification of such pleiotropic genes would be consistent with prior theories that genetic variants encode risk for schizophrenia by causal effects on intermediate phenotypes or endophenotypes of brain structure[9]. We recognised that pleiotropic association per se does not resolve the question of causality, and that the theoretically privileged axis—from gene to brain to schizophrenia—is not the only plausible causal pathway between these entities[10]. However, given recently available statistically well-powered GWAS of schizophrenia and brain

[1]Department of Psychiatry, University of Cambridge, Cambridge, UK. [2]Department of Psychology, University of Cambridge, Cambridge, UK. [3]Department of Genetics and the Neuroscience Center, University of North Carolina at Chapel Hill, Chapel Hill, NC, USA. [4]Cambridgeshire & Peterborough NHS Foundation Trust, Cambridge, UK. [5]These authors contributed equally: Varun Warrier, Edward T. Bullmore. ✉e-mail: ems206@cam.ac.uk

structure, we reasoned that if we could not find any evidence for pleiotropic association, then the role of macro-scale brain structure in mediating schizophrenia risk must be more modest than previously anticipated[11,12].

There is a wide and increasingly diverse range of brain phenotypes that can be measured by MRI or diffusion-weighted imaging (DWI). Most prior large-scale genetic MRI studies have used T1-weighted data to estimate macro-structural phenotypes such as surface area (SA), cortical thickness (CT), and volume[1], each estimated at multiple cortical areas or globally. However, micro-structural MRI metrics are increasingly recognised to provide important additional information about cortical myelination, lamination and other tissue properties, e.g., the density of axons and dendrites measured by neurite density index (NDI)[13]. Both macro- and micro-structural MRI metrics are heritable and can be organised into latent dimensions of shared and distinctive genetic effects[2]. We therefore considered it important to measure genetic associations separately for three minimally correlated MRI metrics (CT, SA, NDI) that emerge developmentally by distinct cellular mechanisms[2]. Recent MRI GWAS data[1,2,14] have shown that single nucleotide polymorphisms (SNPs) often have anatomically localised associations with brain phenotypes, so we considered it important to measure genome-wide association with multiple (3) MRI metrics at each of 180 cortical areas, defined by an anatomically refined prior parcellation template[15], to estimate a set of 540 regional brain phenotypes per scan.

Recognising that the cortex is organised as a complex network[16], we also considered it important to investigate genetic associations with brain network phenotypes. For example, hubs or highly connected nodes have been demonstrated across a wide range of scales and species of nervous systems, including the neuronal network of C. elegans and axonal tract-tracing connectomes of mouse, rat and non-human primate brains[17–20]. In a structural covariance network derived from human MRI[21], hubs typically represent regions that covary strongly with multiple other cortical areas on some MRI metric, e.g., volume or thickness. This phenotypic covariance can be interpreted biologically as a proxy for axonal connectivity and/or shared neurodevelopmental trajectories between strongly covarying regions[21]; and twin studies have shown that human MRI network hubs are heritable[22]. Following Cheverud's conjecture, that phenotypic correlations mirror genetic correlations[23], we expected that structural covariance and genetic similarity would be strongly coupled for each possible pair of cortical areas, and that hubs in the structural covariance network should also be hubs in the genetic correlation matrix. Multiple case-control MRI studies have demonstrated that the normative hubs of structural MRI networks are most locally and topologically atypical in schizophrenia[24,25]. We, therefore, also expected that genes pleiotropically associated with both schizophrenia and brain structure would express their strongest effects on brain regions constituting the hubs of the structural covariance network or connectome.

Previous studies have estimated the genetic correlation between brain and clinical phenotypes, or regressed polygenic scores (PGS) for schizophrenia on brain MRI phenotypes[1,5,8,26]. However, PGS and genetic correlations are unable to pinpoint specific genes or biological processes that are implicated in both brain structure and schizophrenia, as they are composite measures of a large number of SNPs across the genome[7]. To identify specific genes and shared biological mechanisms, the genetic relationship between brain structure and schizophrenia can be more directly investigated by mapping SNPs to genes. We therefore accessed recently published GWAS statistics for three MRI metrics (CT, SA and NDI), each measured at 180 regions in $N = 36,843$ scans from the UK Biobank and the ABCD cohort[2]; and summary statistics from the largest and most recent GWAS study of schizophrenia ($N = 69,369$ individuals with a schizophrenia diagnosis and $N = 236,642$ individuals without a schizophrenia diagnosis)[27]. We mapped each set of SNP-level GWAS statistics to a total of 18,640

individual genes using H-MAGMA[28,29], which takes into account that non-coding SNPs can regulate distal genes via chromatin interaction profiles (measured by high-throughput chromosome conformation capture, Hi-C).

On this basis, we addressed 3 primary questions: (i) is there evidence for pleiotropy between brain (MRI) regional phenotypes and schizophrenia? (ii) is the genetic covariation between brain regional phenotypes and schizophrenia related to brain network phenotypes? and (iii) what are the characteristics of the most strongly pleiotropic genes? These results prompted us to address two more (secondary) questions: (iv) how does schizophrenia compare to other brain disorders in terms of its shared genetic risk with brain structure? and (v) given this pleiotropy, is there a causal pathway for brain-mediated genetic risk of schizophrenia?

## Results
### Genetic associations with regional brain phenotypes
**Identification of genes associated with MRI metrics.** Across all 180 cortical areas, we identified 4222 significant gene-level associations for SA, 773 for CT and 301 for NDI, with false discovery rate (FDR) set at 5% to control type 1 error for multiple comparisons (Methods). For each MRI metric, the number of significantly associated genes varied between cortical regions (Fig. 1A). However, most of the genes were significantly associated with multiple cortical regions, suggesting that shared genetic factors influenced variation of each MRI metric across the cortex, which is in line with high genetic correlations between cortical regions previously reported[1,2]. Aggregating associated genes across all regions, we identified 318 genes in total for SA (i.e., 4222 significant gene-region associations represented 318 non-redundant genes), 157 genes for CT and 86 genes for NDI (Table S1).

Most of these genes were associated specifically with one of the three MRI metrics investigated: 246 out of 318 genes (78%) associated with SA were associated only with SA; 95 out of 157 genes (61%) were associated only with CT; and 43 out of 86 genes (63%) were associated only with NDI. This parallels the minimal genetic correlations across these three MRI metrics[2]. However, 27 genes were significantly associated with all MRI metrics, including 16 genes within the 17q21.31 region. The remaining (11) genes associated with all 3 MRI metrics were located on chromosome 8p23 (7 genes), chromosome 6q25 (3 genes), and chromosome 1p33 (1 gene) (Fig. 1B, Table S2).

**Pathway enrichment and developmental analysis of genes associated with MRI metrics.** The set of 318 genes associated with SA across multiple cortical areas was enriched for 34 Gene Ontology (GO) terms, and the set of 157 genes associated with CT was enriched for 12 GO terms (Methods). SA-related genes were enriched in processes related to central nervous system development (SA GO:0007417, $P = 0.01$) and neurogenesis (SA GO:0050767, $P \leq 0.05$). CT-related genes were enriched for neuron development (CT GO:0048666, $P = 0.01$), neuron projection (CT GO:0031175, $P = 0.01$) and microtubule related processes (CT, GO:0007017, $P \leq 0.002$). Both SA- and CT-related gene sets were enriched for fundamental biological processes such as cell development (SA GO:0048869, $P = 0.01$, CT GO:0048468, $P = 0.05$) (Table S3). The 86 genes associated with NDI were not significantly enriched for any biological pathways.

Using spatio-temporal gene expression data from the PsychEncode database[30], we investigated the developmental expression profiles of gene sets significantly associated with each MRI metric (Methods). Genes associated with CT and SA showed similar trajectories with peak expression during the mid-gestation period (developmental stage 4, 19–22 post-conception weeks (PCW)) followed by a steep decline of expression post-natally. Expression of genes associated with NDI peaked later, in the peri-natal period (developmental stage 5, 35 PCW - 4 months), and gradually decreased post-natally (Fig. 1C; Table S4).

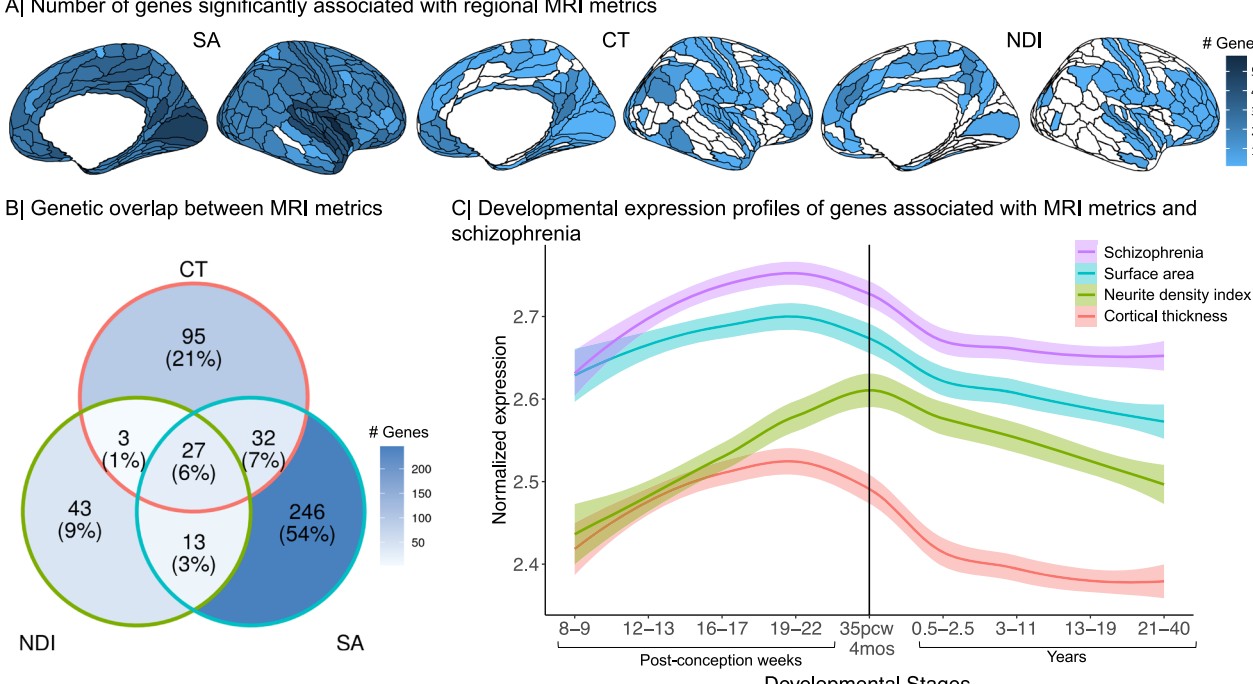

**Fig. 1 | Genetic associations with three MRI metrics of regional brain structure: surface area (SA), cortical thickness (CT) and neurite density index (NDI).**
**A** Cortical surface maps representing the number of genes significantly associated with variation in each MRI metric at each of 180 cortical areas, from left to right: SA, CT, NDI. Regions without any significant gene associations are shown in white.
**B** Venn diagram representing the number of genes that are specifically associated with each MRI metric or generically associated with two or three metrics. The percentages refer to the proportion of all genes associated with one or more MRI metrics represented in each segment of the Venn diagram. **C** Developmental trajectories of average gene expression from 8 post-conception weeks (PCW) to 40 years for the sets of genes significantly associated with each MRI metric or with schizophrenia (SCZ). The shaded region indicates 95% confidence intervals. The vertical line indicates the usual timing of birth. These results highlight mid-to-late fetal stages as a critical window for genetically controlled development of cortical regions[2] and for expression of genes associated with risk of schizophrenia. Source data are provided as a Source Data file.

**Genes associated with schizophrenia and their intersection with MRI-associated genes.** We compared the MRI-associated gene sets with genes identified as significant based on H-MAGMA analysis of an independent GWAS of schizophrenia[27]. We found 587 genes were significantly associated with schizophrenia after correction for multiple comparisons (Table S5) (Methods). In line with previous findings[27,28], this gene set was enriched for 66 GO terms related to neuronal function including nervous system development (GO:0007399, $P \leq 0.001$), neurogenesis (GO:0022008, $P \leq 0.001$) and trans-synaptic transmission (GO:0099537, $P \leq 0.01$) (Table S6). Genes associated with schizophrenia had peak expression during mid-gestation followed by decreasing expression post-natally (Fig. 1B; Table S4)[28].

Out of these 587 schizophrenia-associated genes, 51 were also associated with SA, 22 with CT and 14 with NDI, representing a significant overlap for each metric (SA, $Z = 16.5$; CT, $Z = 8.91$; NDI $Z = 8.14$; all $P \leq 0.0001$ by permutation tests; see Table S7 for details). The genomic region of chromosome 3p21.1 contained several genes pleiotropically associated with both SA and schizophrenia, including PBRM1, NEK4, GNL3, ITIH4 and NISCH. Likewise, the genomic region of chromosome 17q21.31 also contained multiple genes that were associated both with schizophrenia and with all 3 MRI metrics, including NSF, KANSL1, CRHR1, ARHGAP27, LRRC37A, CCDC43, FMNL1, SPPL2C, MAPT, PLEKHM1, STH, and LINC002210-CRHR1. To ensure that pleiotropic association with schizophrenia and MRI metrics was not driven by linkage disequilibrium (LD) between genomic variants, we performed gene set enrichment analysis using MAGMA, which accounts for LD between genes (Methods)[31]. We found that the genetic effects of schizophrenia were enriched for genes significantly associated with each MRI metric (SA, $P \leq 0.0001$; CT, $P \leq 0.001$; NDI, $P \leq 0.01$); and, vice versa, that genes associated with MRI phenotypes were enriched for schizophrenia-related genes (SA, $P \leq 0.0001$; CT, $P \leq 0.001$; NDI, $P \leq 0.05$). These results provide confidence that the evidence for pleiotropic association is not simply driven by LD.

Whilst H-MAGMA aggregates genome-wide SNP level data into gene-level data, an alternative method is to identify genes by fine-mapping significant loci, and linking these to genes, as has been used previously for both schizophrenia[27] and global MRI phenotypes[2]. As a sensitivity analysis, we also investigated if genes prioritised from fine-mapping analyses of schizophrenia (number of genes, $N_G = 106$)[27] are enriched for genes identified from fine-mapping of global MRI metrics (SA, $N_G = 16$; CT, $N_G = 12$; NDI, $N_G = 2$)[2]. The gene effects we identified for SA and CT were enriched for the list of prioritised schizophrenia risk genes ($N_G = 106$)[27], and we replicated the genetic intersection between schizophrenia and both SA and CT metrics that was located on chromosome 17q21.31 using the previously published fine-mapped gene lists[2,27] (see SI Results 2).

**Genetic covariation between schizophrenia and regional MRI metrics**
Since the number of significantly associated genes varied between cortical regions and MRI metrics, and because the genetic covariation between schizophrenia and brain structure might be influenced by genes that do not reach genome-wide significance, we further investigated the genetic relationship between brain structure and schizophrenia without pre-selecting genes based on a $P$-value threshold.

Using partial least squares regression (PLS)[32] to map the anatomical distribution of genetic covariation between regional brain phenotypes and schizophrenia (Fig. 2A, Methods), we found that the first PLS component (PLS1) identified a modest but significant proportion of the genetically determined variation in each MRI metric that covaried with

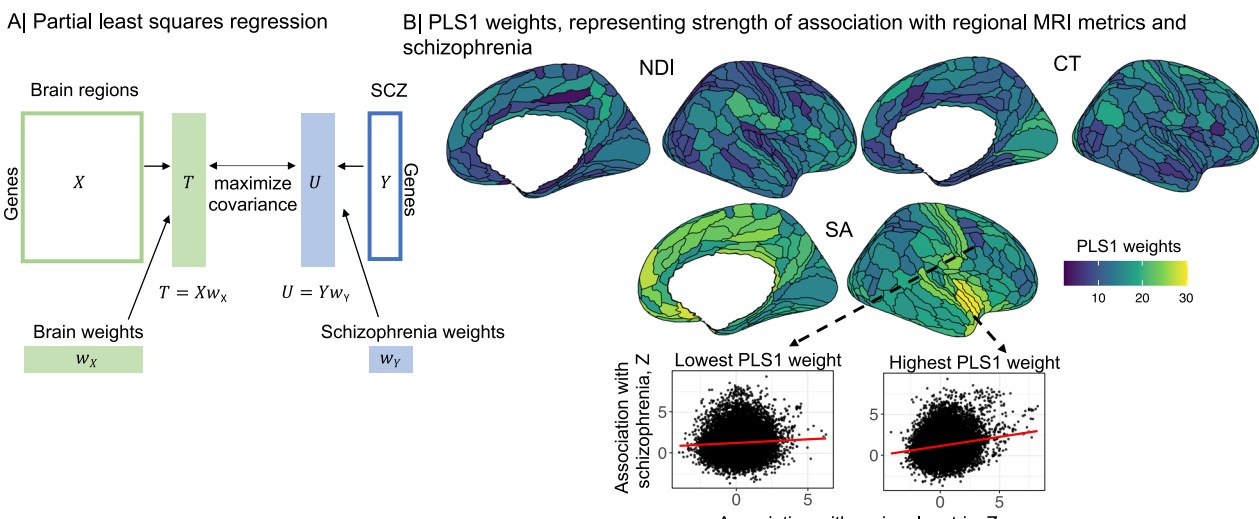

**Fig. 2 | Partial least squares (PLS) analysis of genetic covariation between regional brain phenotypes and schizophrenia. A** The {1 × 18,640 } vector of unthresholded gene association statistics (*Z*-scores) derived by H-MAGMA analysis of the schizophrenia GWAS dataset was designated as the response variable, i.e., the dependent *Y* vector; and the {180 × 18,640} matrix of unthresholded gene association *Z*-scores for each of the MRI GWAS datasets was designated the predictor variable, i.e., the independent *X* matrix. The first PLS component (PLS1) defined the weighted functions of *X* and *Y* that were most strongly correlated overall weighted functions of the whole genome. The PLS1 weights for *X* (brain weights, $w(X)_i$, *i* = 1, 2, 3, ...180) multiplied by *X* constituted a {1 × 18,640} vector of *T* scores (genes weighted by association with brain phenotypes); whereas, the PLS1 weights for *Y* (schizophrenia weights $w(Y)_i$) multiplied by *Y* constituted a {1 × 18,640} vector of *U* scores (genes weighted by association with schizophrenia). Thus genes with the highest absolute *T* and *U* scores can be regarded as the genes which contribute

most strongly to the genetic covariation between schizophrenia and each regional brain phenotype[32,108]. **B** Cortical surface maps of PLS1 weights for neurite density index (NDI), cortical thickness (CT) and surface area (SA). Cortical regions with higher PLS1 weights (shades of yellow) have stronger genetic covariation with schizophrenia: for SA, regions of insular and medial prefrontal cortex; for CT, visual, premotor and inferior parietal cortex; and for NDI, inferior frontal, inferior parietal, posterior cingulate and posterior opercular cortex. Scatterplots (Spearman's correlations, *ρ*) illustrate the genetic relationships between schizophrenia (*y*-axis, *Z*-scores from H-MAGMA analysis of schizophrenia GWAS dataset) and brain surface area (*x*-axis, *Z*-scores from H-MAGMA analysis of MRI GWAS datasets) in two cortical regions, one with a low PLS1 weight (left, dark blue, *ρ* = 0.04), and one with a high PLS1 weight (right, yellow, *ρ* = 0.17). In both plots each point represents one of 18,640 genes. Spearman's correlations were two-tailed. Source data are provided as a Source Data file.

genetic risks for schizophrenia: 5.9% for SA, 5.5% for CT and 3% for NDI (all $P ≤ 0.0001$ by permutation tests). PLS1 weights for each (*i*th) brain region, $w(X)_i$, were *Z*-transformed by bootstrapped standard errors and tested for statistical significance with FDR = 5%.

For SA and CT, all (180) cortical areas had significantly non-zero PLS1 weights, and likewise for NDI at 179 areas, indicating that all MRI metrics were genetically covariant with schizophrenia across large areas of the cortex (Fig. 2B). However, there was substantial regional variation of PLS1 weights for each MRI metric (Fig. 2B). The cortical maps of PLS1 weights were only weakly correlated between MRI metrics (Spearman's $ρ(SA, CT) = 0.03$, $P = 0.65$; $ρ(SA, NDI) = 0.17$, $P = 0.03$; and $ρ(CT, NDI) = 0.19$, $P = 0.01$). In general, however, regions with higher (positive) PLS1 weights had stronger positive correlations between *Z*-scores for association with schizophrenia and *Z*-scores for association with regional brain phenotypes, compared to regions with lower (negative) PLS1 weights; see Fig. 2B.

We tested PLS1 weights for enrichment in relation to prior atlases of laminar differentiation[33] or functional MRI networks[34] (Methods). While the magnitude of PLS1 weights was somewhat related to cytoarchitectonically defined classes of cortical areas, we did not find any enrichment related to functional networks (Tables S8–S9, Fig. S7). These results are compatible with the observation that genetic covariation between brain structure and schizophrenia is expressed diffusely across the cortex[5], and with previously published case-control MRI studies reporting widespread cortical abnormalities in schizophrenia[35].

### Genetic similarity and structural covariance networks

We further investigated the genetic architecture of normative brain structure, going beyond the associations of genetic variation with each independently analysed regional phenotype, to consider genetic associations with network phenotypes. Specifically, we analysed the

relationship between inter-regional phenotypic covariance (henceforth structural covariance *SC*) estimated across *N* = 31,780 scans from the UK Biobank, and inter-regional genetic correlation (henceforth genetic similarity *GS*) (Methods, Fig. 3A).

In line with Cheverud's conjecture[23], we found that there was a strong relationship between structural covariance and genetic similarity. The corresponding structural covariance (*SC*) and genetic similarity (*GS*) matrices for each MRI metric were highly positively correlated with each other, indicating that regions with high structural covariance had highly similar genetic profiles of association with regional variation: for SA, $R(SC, GS) = 0.96$; for CT $R(SC, GS) = 0.93$; and for NDI $R(SC, GS) = 0.94$ (Fig. 3B). Unsurprisingly, *SC* and *GS* matrices of the same MRI metric had consistent relationships with reference atlases of functional network organisation[36] and cytoarchitecture[33] (Figs. S3–S4), Tables S10–S11, Methods). Both structural covariance and genetic similarity were greatest between regional nodes separated by the shortest geodesic distances, and both declined monotonically as a function of increasing distance (Fig. 3C, Fig. S1). However, coupling between *GS* and *SC* remained strong even after controlling for the potentially confounding effects of geodesic distance by regression (for SA, $R(SC, GS) = 0.94$; for CT, $R(SC, GS) = 0.92$; and for NDI, $R(SC, GS) = 0.93$; all $P ≤ 0.0001$) (Fig. 2).

Each of the *GS* and *SC* matrices was represented as a dendrogram by hierarchical cluster analysis and the complex branching structures of the dendrograms were compared in terms of their cophenetic correlation $R_c(GS, SC)$ (Methods). For each MRI metric there was a significantly positive cophenetic correlation between the corresponding *SC* and *GS* dendrograms: for SA, $R_c(GS, SC) = 0.78$; for CT, $R_c(GS, SC) = 0.62$; for NDI $R_c(GS, SC) = 0.46$; all $P ≤ 0.0001$, by permutation tests. This high level of correspondence between the hierarchical community structure of *GS* and *SC* matrices representing the

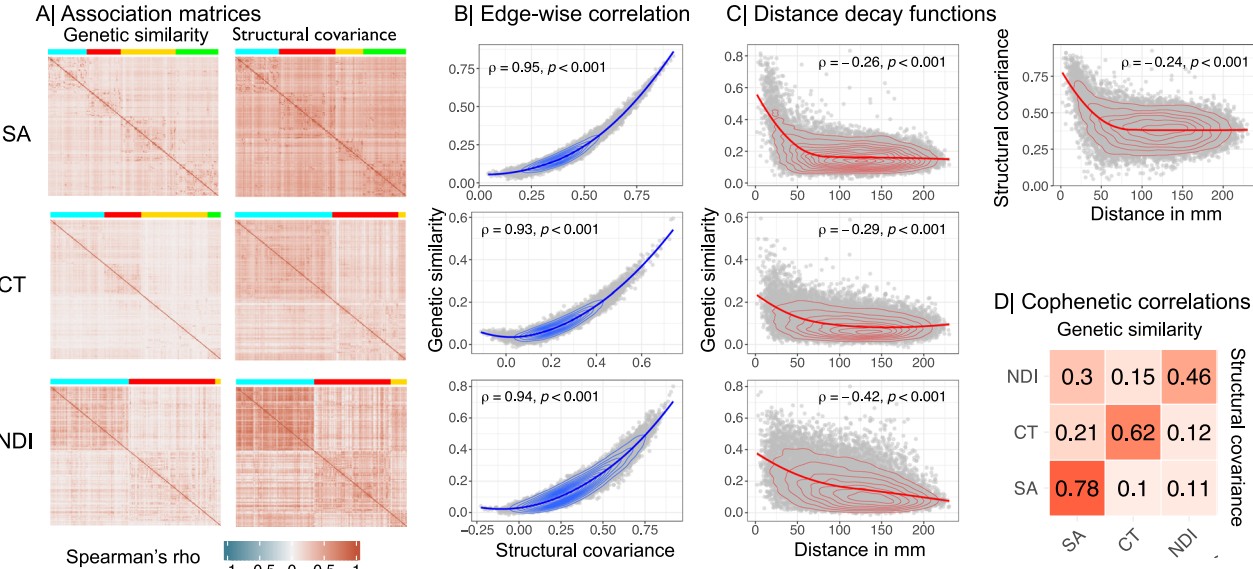

**Fig. 3 | Genetic similarity and structural covariance of cortical networks.**
**A** Genetic similarity (left) and structural covariance (right) matrices for surface area (SA), cortical thickness (CT), and neurite density index (NDI). Brain regions are ordered according to modular decomposition of each matrix; see Fig. 4. **B** Edge-wise Spearman's correlation between genetic similarity ($y$-axis) and structural covariance ($x$-axis) matrices. **C** Spearman's correlation between genetic similarity ($y$-axis) and geodesic distance in millimetres ($x$-axis). For SA, the correlation between structural covariance and geodesic distance is also shown in the top right panel. For genetic similarity, the correlations with geodesic distance were: SA, $\rho = -0.26$; CT, $\rho = -0.29$; NDI, $\rho = -0.42$; all $P \le 0.0001$. Whereas, for structural covariance, the correlations with geodesic distance were: SA $\rho = -0.24$; CT $\rho = -0.3$;

NDI $\rho = -0.4$; all $P \le 0.0001$. Spearman's correlations were two-tailed. **D** Cophenetic correlation matrix showing the similarity in hierarchical clustering of structural covariance and genetic similarity matrices. The upper triangle shows cophenetic correlations based on genetic similarity, the lower triangle is based on structural covariance, and the diagonal represents the similarity between dendrograms of structural covariance and genetic similarity of the same MRI metric. These results indicate that the hierarchical clustering of structural covariance and genetic similarity networks is strongly coupled for each MRI metric, and quite specifically organised for each of the MRI metrics. Source data are provided as a Source Data file.

same MRI metric was greater than the level of correspondence between dendrograms from pairs of *SC* matrices, or pairs of *GS* matrices, representing different MRI metrics (Fig. 3D).

## Genetic relationships between schizophrenia and brain network phenotypes

Since structural MRI covariance is normatively coupled to genetic similarity, and cortical MRI variance is associated with gene variants that are pleiotropically associated with schizophrenia, we predicted that brain regions associated with schizophrenia genes might have distinctive topological profiles in the whole brain connectome represented by the structural covariance matrix. Thus, we tested if the PLS1 weights representing strength of pleiotropic association with schizophrenia-related genes at each cortical region, $w(X)_i$, were related to the weighted degree centrality, $k_i$, or "hubness" of the corresponding node in the *SC* network (Methods).

For each MRI metric, the degree of each node in the corresponding *GS* and *SC* networks was very highly positively correlated (Fig. 4C), indicating that the anatomical distribution of hubs in the structural covariance network closely conforms to the distribution of hubs in the genetic similarity network. This coupling between phenotypic and genetic networks was specific to each MRI metric with much weaker correlation of nodal degrees between *SC* and *GS* matrices derived from different MRI metrics (Fig. 4B).

Degree centrality of nodes in structural covariance networks ("hubness") was positively correlated with regional PLS1 scores, representing the strength of pleiotropic association at each cortical area: SA $\rho = 0.73$, CT $\rho = 0.55$, NDI $\rho = 0.4$; all $P \le 0.0001$ (Fig. 3D). We found that PLS1 scores were also positively correlated with intra-modular and inter-modular degree of each cortical node: intra-modular degree SA $\rho = 0.55$, CT $\rho = 0.48$, NDI $\rho = 0.22$; inter-modular degree SA $\rho = 0.39$, CT $\rho = 0.38$, NDI $\rho = 0.73$; all $P \le 0.0001$ (Fig. S6A, B).

## Pleiotropic genes mediating covariation between schizophrenia and regional MRI metrics

To identify individual genes which made the greatest contribution to whole genome covariation between schizophrenia and regional brain MRI metrics, we focused on the $T$ and $U$ scores derived from PLS analysis (Methods and Fig. 2), and the correlation between them, $R(T, U)$.

The strength of pleiotropic association with schizophrenia, across all 18,640 genes, was greater for SA ($R(T, U) = 0.24$), than for CT ($R(T, U) = 0.23$), or NDI ($R(T, U) = 0.17$); see Fig. 5A. To assess the influence of an individual gene on these whole genome relationships, we used a leave-one-out (LOO) strategy and computed $\Delta(R(T, U))$, i.e., the difference between the original and the LOO $R(T, U)$, for each gene. Genes that make the greatest individual contribution to pleiotropic association will have the largest positive values of $\Delta(R(T, U))$ (Methods).

We functionally characterised the top 1% of genes with largest positive $\Delta(R(T, U))$ for each MRI metric (Methods). These sets of 185 genes were significantly enriched for constrained genes, which are intolerant of damaging variants and have previously been associated with brain structure[2] and schizophrenia[37] (Fig. S9A). The pleiotropic gene sets were not strongly enriched for cell type-specific genes (Fig. S9B, Table S15) but we identified 33 significant GO enrichments (six for SA, 22 for CT and five for NDI) (Table S16) related to neurodevelopmental processes including neurogenesis (SA GO:0022008, $P = 0.001$), nervous system development (CT GO:0007399, $P = 0.002$), glial cell development (CT GO:0048468, $P = 0.009$) and neuron projection development (NDI GO:0010975, $P = 0.03$). In a sensitivity analysis, we observed a similar pattern of enrichment for the top 3% ($N_G = 556$) of genes with largest positive $\Delta(R(T, U))$ (Fig. S9A, B, Tables S14–S15).

We tested whether pleiotropic genes were over-represented in discrete genomic regions using hypergeometric testing implemented

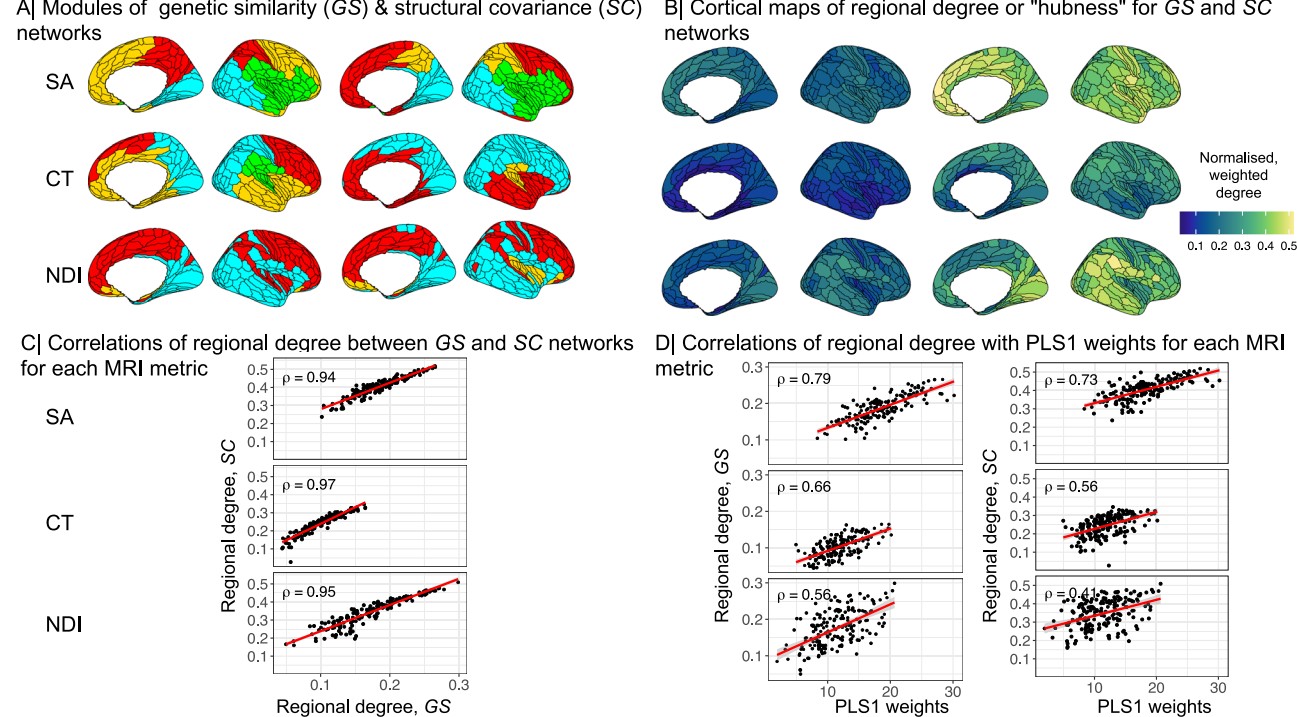

**Fig. 4 | Hubs of genetic similarity and structural covariance networks are co-located and associated with pleiotropic genes. A** Modular decomposition of genetic similarity matrices (left) and structural covariance matrices (right) for surface area (SA), cortical thickness (CT) and neurite density index (NDI). We used the Louvain algorithm to resolve the modular community structure of *SC* and *GS* networks for each MRI metric and found three (for NDI) or four (for CT, SA) spatially contiguous modules of the *GS* networks, and three (CT, NDI) or four (SA) modules of the corresponding *SC* networks (Methods). **B** Cortical surface maps of hub scores based on genetic similarity matrices (left) and structural covariance matrices (right). **C** Scatterplots showing positive Spearman's correlations between hubness (weighted degree centrality) of nodes in genetic similarity (x-axis) and structural covariance networks (y-axis) for each MRI metric. **D** Scatterplots showing positive Spearman's correlations between strength of pleiotropic gene association indexed by PLS1 weights (x-axis) and hub scores of nodes in genetic similarity networks (left) or structural covariance networks (right) (y-axis). The shaded region indicates 95% confidence intervals. Spearman's correlations were two-tailed. Source data are provided as a Source Data file.

in FUMA[38] (Methods). We identified eight loci of significant positional enrichment for one or more of the 3 MRI metrics, most of which were specific to a single metric (Fig. 5B). However, genes pleiotropically associated with all 3 MRI metrics were positionally enriched at the same three genomic regions: chromosome 3p21 (chr3:43,700,001-54,400,000), chromosome 17q21 (chr17:38,100,001-50,200,000) and chromosome 11p11 (chr11:43,500,001-53,700,001). The strongest positional enrichment for each MRI metric was on chromosome 3p21, suggesting that genes concentrated at this location strongly influenced the genetic relationship between schizophrenia and all three brain MRI metrics (Fig. 5C).

Since the loci of positional enrichment identified using FUMA were relatively long (≥5 Mb, ≤25 Mb), and to ensure that positional enrichments are robust to methodological choices, we also estimated local genetic correlations at these positions to identify shorter genomic segments (~1Mb) mediating pleiotropic associations using LAVA (Local Analysis of [co]Variant Association)[39]; see Methods. Within four of the positionally enriched loci identified by FUMA, we were able to resolve the pleiotropic association to smaller subregions (≥1 Mb, ≤2 Mb), i.e., chromosomal loci 14q32.2–14q32.31 and 14q32.13 for CT, and 3p21.2–3p21.1, 2q33.1 and 17q21.31 for SA (Fig. 5B). We did not find any significant local genetic correlations for NDI (Table S16).

**Clinical diagnostic specificity of genetic covariation between schizophrenia and brain structure**
We repeated many of the principal analyses of pleiotropic association with schizophrenia, using identical methods and models applied to independent large-scale GWAS data for two additional neuropsychiatric disorders−bipolar disorder (BIP) (N = 41,917 cases and N = 371,549

controls)[40] and Alzheimer's disease (AD) (N = 398,058)[41]−and for height, a neurodevelopmentally sensitive non-psychiatric phenotype (N = 4,080,687)[42].

We identified genes significantly associated with BIP, AD or height and investigated their intersection with MRI-associated genes. Out of 136 BIP-associated genes, only the intersection with 15 genes also associated with SA was significant, and largely comprised genes located at chromosome 3p21. Out of 77 AD-associated genes, there were significant intersections with genes also associated with SA (15), CT (12), and NDI (11); and several genes associated with both AD and surface area were located at chromosome 17q21. Out of 8012 genes associated with height, 175 were associated with SA, 55 with CT and 23 with NDI; 21 genes were shared between height and all 3 MRI metrics, and these were located on chromosome 17q21, 8p23 and chromosome 1p33.

We also used PLS regression, as previously for analysis of whole-genome covariation with schizophrenia (Fig. 2), for comparable analysis of regional brain phenotypes that were pleiotropically associated with risk of BIP, AD or height; see Fig. 6 and SI Results 6 for details. The proportion of disorder-related variance explained was greater for schizophrenia (SCZ: SA = 5.9%, CT = 5.5%, NDI = 3%), than for bipolar disorder (BIP; SA = 2.7%, CT = 3.6%, NDI = 1.6%), or Alzheimer's disease (AD; SA = 1.5%, CT = 1.2%, NDI = 0.8%). The proportion of height-related variance was comparable to the proportion of schizophrenia-related variance across all MRI metrics (Height; SA = 7.1%, CT = 5.5%, NDI = 2.7%). The proportion of overlapping genes was less than 50% for bipolar disorder and schizophrenia and further decreased for either Alzheimer's disease or height and schizophrenia. This implies that the genes identified for MRI metrics and schizophrenia are largely distinct from the genes identified for Alzheimer's disease or height.

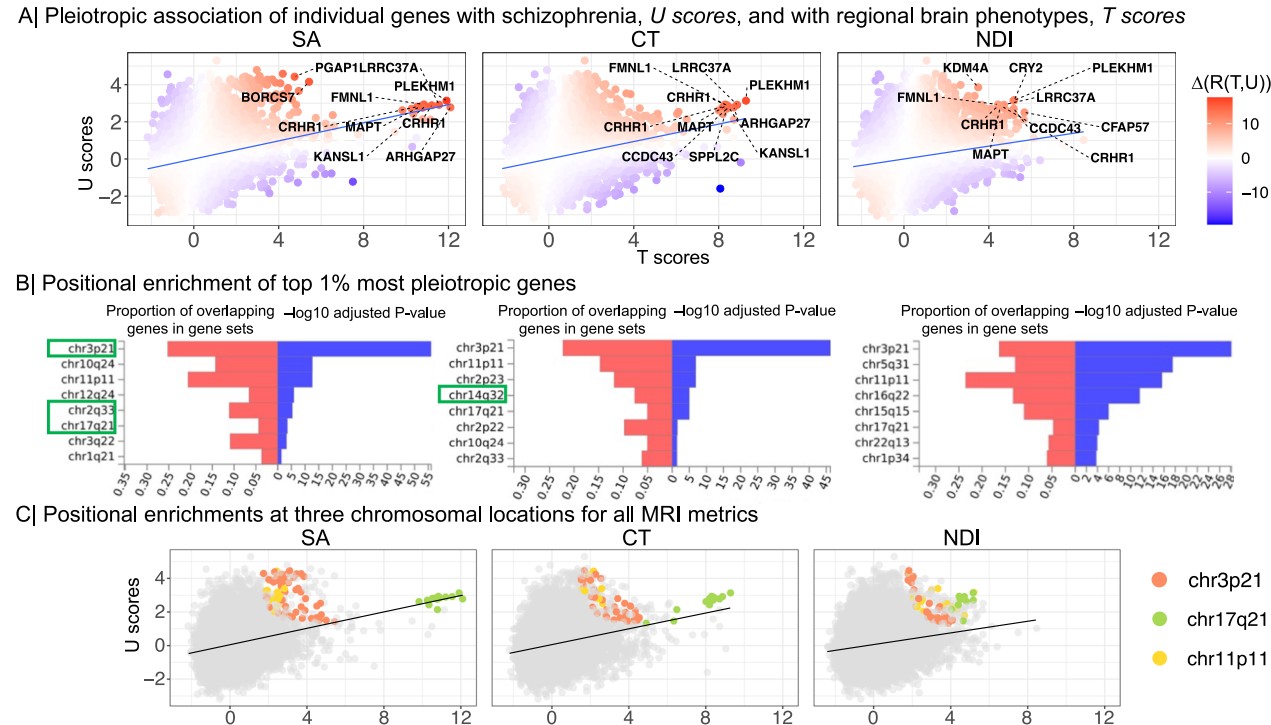

**Fig. 5 | Genes pleiotropically associated with schizophrenia and regional MRI metrics. A** Scatterplot of *T* scores (*x*-axis) versus *U* scores (*y*-axis) for each of 18,640 protein-coding genes, derived from their PLS1 weights (Fig. 2). The *T* score is the weight of each gene on the MRI metric; the *U* score is the weight of each gene on the association with schizophrenia; and the correlation between *T* and *U* scores, *R*(*T*, *U*), quantifies the strength of genetic relationship between brain and schizophrenia phenotypes. Each gene is colour-coded according to its value of Δ(*R*(*T*, *U*)) which indicates the positive (red) or negative (blue) magnitude of its influence on the whole genome relationship between schizophrenia and each MRI metric. The top ten genes with the largest positive leave-one-out scores for Δ(*R*(*T*, *U*)) are annotated, including PLEKHM1, FMNL1, LRRC37A, MAPT, KANSL and CRHR1, all located within the 17q21.31 region. We note that these genes were also identified by the intersection analysis of genes significantly associated with both MRI metrics

and schizophrenia (Fig. 1). **B** Significant positional enrichment of 185 genes (top 1%) with the highest Δ(*R*(*T*, *U*)) scores based on hypergeometric testing implemented in FUMA. For example, the genes most strongly contributing to genetic covariation between SA and schizophrenia were positionally enriched at chromosome 2q33, whereas genes contributing to covariation between CT and schizophrenia were enriched at chromosome 14q32. The red bars show the proportion of co-located genes according to the size of each gene-set; the blue bars indicate -log10 *P*-values adjusted for the number of tested gene-sets. Chromosomal locations showing significant local genetic correlations based on LAVA are highlighted in green boxes. **C** Scatterplot of *T* scores versus *U* scores, exactly as shown in (**A**) except that genes are colour-coded according to their location in the three genomic regions that were positionally enriched for all MRI metrics. Source data are provided as a Source Data file.

## Causal relationships between brain and schizophrenia phenotypes

We used two-sample Mendelian randomization (MR) analysis to test two directions of causal relationship between brain and schizophrenia phenotypes: (i) schizophrenia (exposure) causing brain changes (outcome); and (ii) brain changes (exposure) causing schizophrenia (outcome) (Methods). We restricted Mendelian randomization analysis to a subset of regional MRI metrics that showed ≥5 genome-wide significant loci, to ensure reasonable statistical power[43]. This condition was not satisfied for all cortical areas by any MRI metric: out of 180 regions, 48 had ≥5 gene-level associations with SA, but there were only 10 regional NDI phenotypes and 5 regional CT phenotypes which passed the criterion. After correcting for multiple comparisons with FDR 5%, we did not find any significant evidence for a causal effect of schizophrenia on the cortical thickness, surface area or NDI of this subset of brain regions. However, there was evidence for a significant causal effect of genetically predicted brain structure on schizophrenia (Fig. S11). Specifically, SA of V4 and ProS cortical areas was predictive of risk for schizophrenia (inverse variance weighted method: V4, $\beta = 0.38$, $SE = 0.1$, $P = 0.02$; ProS, $\beta = 0.26$, $SE = 0.05$, $P = 0.0002$). For ProS (prostriate cortex), a region of posterior cingulate cortex, sensitivity analyses indicated that the effect of this exposure on the outcome of schizophrenia was robust and not attributable to horizontal pleiotropy. For V4, a region of ventral occipital cortex specialised for

colour vision, sensitivity analyses were less consistent and indicated potential horizontal pleiotropy (See SI Results 7).

## Discussion

To address our first question (i), about the evidence for pleiotropic association with schizophrenia and brain regional phenotypes, we began by identifying genes associated with one or more of three MRI metrics at one or more of 180 cortical regions in an adult general population cohort. Most genes associated with any brain phenotype were only associated with one phenotype, indicating some specificity of genetic effects for distinct MRI metrics. This must be caveated by the limited number of MRI metrics considered, and the relatively small sample sizes currently available for GWAS of any brain MRI phenotypes. However, genetic associations specific to different MRI metrics are compatible with previous studies demonstrating that CT, SA and NDI are genetically distinct[1,2]. Genes associated with each MRI metric broadly shared transcriptional trajectories that peaked during mid-late periods of fetal life, which is well-recognised as a key neurodevelopmental period for the formation of upper layer neurons and neuronal differentiation (including axonogenesis and dendritic arborization)[44]. A minority of genes were generically associated with all three MRI metrics, including 16 genes located within the 17q21.31 region. Genes associated with SA and CT were enriched for neurodevelopmental processes including microtubule function, which is known to be important for

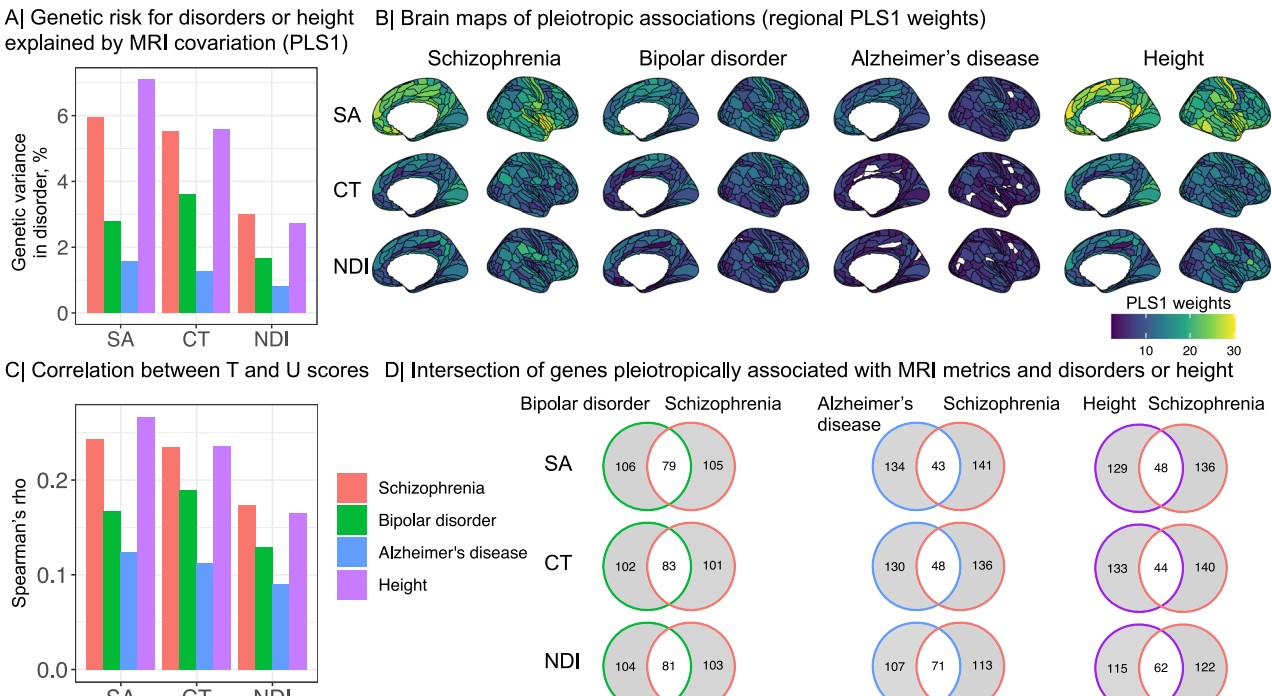

**Fig. 6 | Specificity of pleiotropic associations between clinical disorders or height and regional brain phenotypes. A** Proportion of variance in the genetically predicted risk for each disorder and height (*y*-axis) explained by the genetic effects on regional MRI metrics (*x*-axis; SA surface area, CT cortical thickness, NDI neurite density index) based on the first PLS component, PLS1. **B** Cortical surface maps of PLS1 regional brain weights for schizophrenia (SCZ), BIP, AD and height. Higher positive weights (shades of yellow) indicate stronger genetic covariation with each disorder; regions with zero weight are shown in white. Mean absolute weights were lower for BIP (SA $\bar{w} = 12.3$, CT $\bar{w} = 9.45$, NDI $\bar{w} = 8$), and for AD (SA $\bar{w} = 9$, CT $\bar{w} = 5.2$, NDI $\bar{w} = 5.4$), than for schizophrenia (SA $\bar{w} = 18.29$, CT $\bar{w} = 11.89$, NDI $\bar{w} = 11.37$). Apart from SA, mean PLS weights for height were generally lower than for schizophrenia (SA $\bar{w} = 20.4$, CT $\bar{w} = 12.1$, NDI $\bar{w} = 10.5$). Fewer brain regions had significant PLS1 scores for BIP (NDI = 175) and AD (SA = 79, CT = 166, NDI = 170) than for schizophrenia (SA, CT = 180, NDI = 179). For height, all brain regions showed significant PLS1 scores. **C** Spearman's correlations ($\rho$; *y*-axis) between T and U scores for schizophrenia, bipolar disorder, Alzheimer's disease and height. The strength of pleiotropic association indexed by $\rho$ was greater for schizophrenia (SA $\rho = 0.24$, CT $\rho = 0.23$, NDI $\rho = 0.17$), than for BIP (SA $\rho = 0.17$, CT $\rho = 0.19$, NDI $\rho = 0.13$), AD (SA $\rho = 0.12$, CT $\rho = 0.11$, NDI $\rho = 0.09$). For SA, the pleiotropic association with height was stronger compared to schizophrenia (SA $\rho = 0.27$, CT $\rho = 0.23$, NDI $\rho = 0.16$). **D** Venn diagrams showing the intersection of the top 1% most pleiotropic genes, with the highest $\Delta(R(T, U))$ scores, for each MRI metric. Source data are provided as a Source Data file.

neurogenesis, neuronal migration and axon generation[45]. Collectively, these results indicated that genetic associations with brain structural phenotypes likely represent the effects of genetically controlled programmes of gestational and early post-natal development of the cortex.

We then found that there was significantly greater than expected intersection or overlap between the gene sets associated with regional brain phenotypes and the gene set associated with schizophrenia. For example, about 9% of the 586 genes significantly associated with schizophrenia were also significantly associated with normative variation in surface area of one or more cortical regions. We subsequently used partial least squares to investigate genetic covariation over the whole genome and to comprehensively map cortical locations of significant genetic covariation with schizophrenia. We found evidence for significant pleiotropic association with both schizophrenia and all three MRI metrics at all (SA, CT) or almost all (NDI) cortical areas, indicating that genetic risks for schizophrenia are also associated with widespread variations in regional cortical anatomy. This observation is consistent with global or extensive, diffuse regional differences in cortical structure reported by prior case-control MRI studies of schizophrenia[35]; but there was also considerable inter-areal variation in the strength of pleiotropic association for each metric. For example, genetic covariation with surface area was greatest in paralimbic areas of cortex that have previously been associated with polygenic risk for schizophrenia[5]. Overall, we consider that there is strong evidence for pleiotropic association with regional brain phenotypes and schizophrenia.

To address the second question (ii) about the genetic covariation between MRI phenotypes and schizophrenia in relation to brain network phenotypes, we first investigated the concordance between genetic and phenotypic brain networks. For each MRI metric, structural covariance and genetic similarity networks were highly correlated in terms of edge weights, weighted nodal degree, and hierarchically clustered community structure. For example, the hubs of the cortical thickness covariance network were generally also hubs in the genetic similarity network derived from GWAS data on cortical thickness at each node. Genetic similarity network hubs are cortical areas that share a whole genome profile of genetic association in common with many other areas of cortex and are, therefore, putatively under the same genetic controls throughout development. One interpretation of the strong coupling between genetic and structural covariance network hubs is that cortical areas are more likely to be anatomically connected if their differentiation and development is controlled by similar genetic mechanisms. There is convergent evidence in support of this "grow together, wire together" model from studies in animals and humans demonstrating that the probability of inter-areal axonal projections is increased between cortical areas with similar cytoarchitectonics[46] or whole genome transcription profiles[47]. The neurobiological substrate of structural covariance networks has long been debated[21] but our results, in line with Cheverud's conjecture, strongly suggest that structural covariance between two regions of the adult brain represents close equivalence in the genetically determined trajectories of their development since mid-fetal life.

In this normative context, it was notable that the regional brain structural phenotypes with the highest genetic covariation with schizophrenia were the hubs of the corresponding structural covariance networks. It seems plausible that the neurodevelopmentally-enriched genes associated with schizophrenia also have an important role in development of anatomical inter-connectivity between cortical areas, and are therefore most strongly associated with the most highly connected areas of the cortical network. The flip side of this interpretation is that genetic variants associated with schizophrenia may cause atypical development of brain network hubs, in particular, with emergent consequences for "higher order" cognitive processes that are often impaired in schizophrenia[48] and dependent on the integrative aspects of brain network topology mediated by hubs[49]. Consistent with this concentration of pleiotropic effects on hubs of the connectome, previous studies have shown that whole-genome transcriptional profiles co-located with adolescent myelination of structural covariance network hubs were enriched for genes associated with schizophrenia[50]; and that schizophrenia cases, compared to healthy controls, had reduced degree of cortical hubs in morphometric similarity networks that were co-located with cortically patterned expression of schizophrenia-related genes[25]. Overall, we consider that there is strong evidence for pleiotropic association with brain network phenotypes and schizophrenia.

To address our third question (iii), about characterisation of the most pleiotropic genes, we deployed a wide range of enrichment analyses. Across all MRI metrics, we found that the most pleiotropic gene sets were enriched for constrained genes, that are intolerant of damaging variants, echoing prior reports of constrained gene enrichment in the genetic architecture of schizophrenia[27,51]. Pleiotropic genes were also enriched for neurodevelopmental and glial ontology terms, and had peak expression during the second half of normal gestation, consistent with the role of glial cells in typical and atypical development of brain networks[52,53]. These findings are in line with the neurodevelopmental model of schizophrenia, positing that genetic and early environmental factors perturb normal processes of brain development, including atypical formation of synaptic connections and axonal projections, with anatomically distributed effects on adult brain connectivity, that predispose individuals to develop psychotic symptoms later in life[7,9,54].

However, one of the most striking, robust and contextually plausible characteristics of highly pleiotropic genes was their positional enrichment or genomic clustering on three chromosomal regions: chromosome 3p21, chromosome 11p11 and chromosome 17q21. While some of these genomic regions have previously been related to psychiatric disorders or MRI phenotypes, this study clearly implicates these regions by analysing the genetic underpinnings of schizophrenia and MRI metrics simultaneously.

Genes within 17q21 were consistently identified across the different methodological approaches. Encouragingly, genetic variation in the 17q21 region has been replicably associated with various measures of brain structure[55–58], as well as with schizophrenia[27,59], in prior studies. However, chromosome 17q21 has also been associated with other disorders, such as autism spectrum disorder[60], Alzheimer's disease[61], suggesting that this region might have effects on brain phenotypes that contribute to the pathogenesis of several neuropsychiatric diseases. Using local genetic correlation analysis (LAVA), we were able to resolve the pleiotropic locus for surface area to a subregion of 17q21.31 spanning ~1.4 Mb. This locus harbours an inversion polymorphism with a complex LD structure[62] and includes PLEKHM1, MAPT, KANSL1 and CRHR1. CRHR1 encodes the main receptor of corticotrophin-releasing hormone and has recently been highlighted in a study of shared genetic effects on schizophrenia and subcortical volumes[63]. MAPT encodes microtubule-associated protein tau, which is known for its role in axonal transport and neurite outgrowth and has previously been associated with schizophrenia and structural MRI

metrics[6]. PLEKHM1 is involved in autophagy[64], a process that has been suggested to have a key role in the pathophysiology of schizophrenia[65].

Chromosome 11p11 also harbours genes that have been previously associated with schizophrenia and/or brain structural phenotypes, including CHRM4, MDK, AMBRA1 and HARBI1[27,66,67]. For example, CHRM4, encoding the muscarinic acetylcholine receptor M4, has been linked to the genetic risk for schizophrenia, with reduced hippocampal expression in post mortem cases[68], and positive clinical trials of M4 agonists for the treatment of schizophrenia[69,70]. Chromosome 3p21 has also previously been associated with schizophrenia and other psychiatric disorders[27,37,71], cognitive[72] and brain MRI phenotypes[58]. Using LAVA, we resolved the pleiotropic locus to the subregion 3p21.2 - 3p21.1, spanning ~ 2.1 Mb. This region harbours genes including ITIH4, NEK4, GNL3 and PBRM1 that have previously been linked mechanistically to schizophrenia and related brain changes at cellular and whole brain scales[4,71,73].

Overall, these results converge to provide strong evidence for pleiotropic associations of common genetic variants with both brain structure and risk of schizophrenia. However, they raised two secondary questions about specificity and causality. The question of clinical diagnostic specificity (iv) is whether there are similar genetic associations with both brain phenotypes and risk for other neuropsychiatric disorders. We repeated the analysis for pleiotropic associations with regional brain phenotypes, exactly as we had done for schizophrenia, using large GWAS datasets on bipolar disorder, Alzheimer's disease and height. We found that genetic covariation with brain structure was stronger for schizophrenia than for bipolar disorder, Alzheimer's disease or height, and pleiotropic genes were largely specific to each disorder, although there were also some genes and loci that were pleiotropically associated with more than one brain disorder. For example, the chromosome 17q21 locus was associated with cortical surface area and both schizophrenia and Alzheimer's disease (it has previously been linked to AD alone[61]) and height; and the chromosome 3p21 locus was pleiotropically associated with surface area and both schizophrenia and bipolar disorder (it has previously been linked to BIP alone[74]). For SA we found that the genetic covariation was somewhat greater with height compared to its genetic covariation with schizophrenia. This finding is in line with studies reporting phenotypic and genetic correlations between height and surface area[1,75–77], as well as between height and schizophrenia[78,79]. It is also consistent with our finding that height was strongly positively correlated with SA (but not CT or NDI) in these data. This implies that genetic variants located at 17q21 may have normative effects on the correlated phenotypes of height and brain surface area as well as conferring increased risk of schizophrenia. These are plausible but preliminary results and several limitations need to be considered. First, this study was limited to common variants. However, it is known that schizophrenia is additionally associated with rare variants[9]. Second, the PLS analyses were based on a specific SNP-to-gene mapping method (i.e., H-MAGMA). Further studies, also integrating rare variants, and multiple SNP-to-gene mapping methods, will be needed to survey the commonalities and differences between brain disorders in terms of their genetic relationships with brain structure.

The question of causality (v) arises because pleiotropy is a necessary (but not a sufficient) condition for the standard causal model of biological pathogenesis: that genetic variation causes brain changes, which in turn cause schizophrenia[9]. This model was not refuted by lack of evidence for pleiotropic association in this study. However, no amount of evidence for pleiotropic association can resolve the causal relationship between the two genetically coupled phenotypes: do brain phenotypes cause schizophrenia or vice versa? Mendelian randomization provides a potentially powerful approach to address this question more directly and we used it to test both the standard causal pathway— gene → brain → schizophrenia—and the alternative causal pathway— gene → schizophrenia → brain (as previously reported for frontal

cortex[10]). We found no evidence for the alternative pathway but some evidence for the standard pathway. Genetically predicted surface area of two cortical regions (ProS and V4) was predictive of schizophrenia, and the effect of ProS on schizophrenia was robust to sensitivity analyses (see SI 7). The posterior cingulate cortex has previously been associated with polygenic risk scores for schizophrenia[5,80,81] and schizophrenia- and schizotypy-related abnormalities of macro- and micro-structural MRI metrics have previously been reported in this region[82–85]. It seems plausible that genetically determined changes in surface area of the posterior cingulate cortex might cause increased risks of schizophrenia, as suggested by these results. However, the MR analyses that we were able to do were limited. Only a minority of regional brain phenotypes had sufficient, robustly significant genetic instruments for MR analyses. This reflects the relatively small size of currently available MRI GWAS datasets[1,2], e.g., compared to GWAS for schizophrenia[27], which likely constrained our statistical power to detect multiple small-effect gene variants associated with many brain phenotypes[10,86]. For more definitive future investigations of the fundamentally important question of causal relationships between brain structure and brain disorders it will be essential to build larger GWAS datasets of anatomically comprehensive and technically diverse MRI phenotypes.

## Methods

### Inclusion and ethics
Ethical procedures for the UK Biobank are controlled by the Ethics and Guidance Council (http://www.ukbiobank.ac.uk/ethics), and the study was conducted in accordance with the UK Biobank Ethics and Governance Framework document (https://www.ukbiobank.ac.uk/media/0xsbmfmw/egf.pdf), with institutional review board approval by the North West Multi-centre Research Ethics Committee.

### Genome-wide association studies (GWAS)
We accessed recently published GWAS summary statistics on three regional brain phenotypes including surface area, cortical thickness and neurite density index, measured at 180 brain regions (3 x 180 GWAS). The GWAS's were based on 36,843 subjects from the UK Biobank[87] and the Adolescent Brain and Cognitive Development (ABCD) study[88]. Details can be found in ref. 2. To investigate genetic covariation with schizophrenia, we accessed GWAS summary statistics based on 69,369 schizophrenia cases and 236,642 controls[27].

### Mapping GWAS data to genes using H-MAGMA
We performed SNP-to-gene mapping for each GWAS using H-MAGMA[28]. We used H-MAGMA for two reasons: (i) compared to gene-mapping methods that only use positional information, H-MAGMA can incorporate tissue-specific and long-range regulatory effects[28]; and (ii) although alternative transcription-based gene-mapping methods can also integrate gene regulatory information, currently available datasets on expression quantitative trait loci (eQTL) in fetal brain do not have large-enough sample sizes, and do not capture trans-eQTL effects well[89,90]. Additionally, H-MAGMA allows developmental stage-specific gene mapping by integrating chromatin interaction profiles from fetal brain (gestation weeks 17–18) or adult brain (age 36–64 years) Hi-C datasets[28,29].

Single-nucleotide polymorphisms (SNPs) were mapped to genes using the default settings in H-MAGMA with fetal and adult brain Hi-C annotation data files provided by the developers of H-MAGMA (https://github.com/thewonlab/H-MAGMA) and the reference data file for a European ancestry population downloaded from https://ctg.cncr.nl/software/magma. Since the genes associated with both schizophrenia and brain structure were enriched for regulatory regions active in the fetal brain[2,91], we report findings based on fetal Hi-C datasets. We restricted our analyses to protein-coding genes and excluded genes within the major histocompatibility region due to its complex LD structure[28].

## Genetic associations with brain MRI metrics and schizophrenia: identification and characterisation

### Identification of genes associated with MRI metrics.
To investigate the gene-level associations with regional brain structure, we identified a set of significantly associated genes for each MRI metric. To account for multiple comparison correction, we performed matrix decomposition within each MRI metric to identify the number of independent phenotypes ($N_{pheno}$)[92]. Significance thresholds were then calculated for each MRI metric by Bonferroni correction based on the total number of tests ($P = 0.05/18,640$ genes x $N_{pheno}$)[2]. We identified 78 independent phenotypes for SA, 113 for CT and 74 for NDI. The probability thresholds for significant association varied accordingly between MRI metrics (SA $P \le 3.44 \times 10^{-8}$; CT $P \le 2.37 \times 10^{-8}$; NDI $P \le 3.62 \times 10^{-8}$).

### Identification of genes associated with schizophrenia.
To identify genes that were significantly associated with schizophrenia, we mapped GWAS summary statistics based on 69,369 schizophrenia cases and 236,642 controls[27] to genes using H-MAGMA. We restricted genes to protein coding genes and excluded genes located in the major histocompatibility complex (MHC) region. We performed Bonferroni correction for all 18,610 genes leading to a significance threshold of $P \le 2.69 \times 10^{-6}$. We used hypergeometric testing implemented in the R package GeneOverlap[93] to test for significant overlap between schizophrenia-associated genes and MRI metric-associated genes and performed permutation testing (10,000 permutations) to test whether this overlap is non-random. Additionally, we performed gene-set enrichment analysis using MAGMA, which accounts for linkage disequilibrium between genetic variants[31]. First, we tested whether the genetic effects of schizophrenia were enriched for gene-sets significantly associated with each MRI metric. Gene-sets for MRI metrics included all the genes that we identified as significant for each MRI metric (SA $N_G = 318$; CT $N_G = 157$; NDI $N_G = 86$). Second, we tested whether the genetic effects of each MRI metric were enriched in a gene-set significantly associated with risk for schizophrenia ($N_G = 587$).

### Pathway enrichment and neurodevelopmental expression profiling of schizophrenia- and brain-related genes.
We used the R package gProfiler2 to conduct GO enrichment analysis, which resembles gene set enrichment analysis, and focused enrichment tests on biological processes[94]. We investigated developmental expression profiles of genes that were significantly associated with regional brain structure or schizophrenia. To ensure that developmental expression trajectories were not biased by the developmental stage from which Hi-C data were obtained, we created lists of significantly associated genes that were identified using either fetal or adult Hi-C data[28]. A spatio-temporal transcriptomic atlas from PsychEncode[30] was used to estimate cortical expression profiles across nine developmental stages (window 1: post-conceptional week (PCW) 5-9, window 2: PCW 12-13, window 3: PCW 16-18, windows 4: PCW 19-22, window 5: PCW 35- PY 0.3, window 6: post-natal year (PY) 0.5-2.6, window 7: PY 2.8-10.7, window 8: PY 13-19, window 9: PY 21-64). This data set contained expression values from multiple brain regions. Analyses were restricted to the cortex. Expression values were log-transformed and centred to the mean expression level for each sample using the `scale(center = T, scale= F)+1` function in R[95]. To do this, genes significantly associated with regional brain phenotypes were selected for each cortical transcript and averaged over all sampled areas of cortex and their mean-centred expression values were calculated and plotted over time. Additionally, we investigated the effect of developmental window (prenatal expression included windows 1-4, post-natal expression included window 6–9) on gene expression using linear mixed effect models, with a fixed effect of developmental window and gene length, and a random effect of brain region.

## Structural covariance and genetic similarity networks

To generate structural covariance matrices, we accessed imaging data from the UK Biobank and focused on a subset of $N = 40,680$ participants for each of which complete genotype and multimodal MRI data were available for download (February 2020)[87]. Details on processing pipeline and exclusion criteria and are described in SI Methods 1. After quality control, $N = 31,780$ MRI scans were available for all MRI metrics. This sample comprised ~53% female, ~47% male participants aged 40–70 years, with mean age 55 years (SD = 7.4 years) and it was a subset of the sample used for MRI GWAS[2]. For each MRI metric, we estimated the pair-wise correlation of regional phenotypes for all pairs of 180 regions over all $N = 31,780$ MRI scans to constitute a symmetric, signed {180 × 180} structural covariance matrix. Each unique, off-diagonal element of this matrix, $SC_{i,j}$ can be regarded from the perspective of graph theory as the weight of an undirected edge between regional nodes $i$ and $j$ in a whole brain structural connectome. We also estimated the similarity of genetic association between each possible pair of regional nodes $i$ and $j$ in terms of Spearman's correlation, $\rho_{i,j}$, between their gene-level association statistics (Z-scores derived from H-MAGMA) to constitute a symmetric, signed {180 × 180} genetic similarity (GS) matrix.

## Geodesic distance, functional networks and cytoarchitectonic classes.

We investigated the association between genetic similarity (or structural covariance) and geodesic distance, known functional networks and cytoarchitectonic classes. Geodesic distance was defined as the length between each pair of regions along the cortex measured in millimetres and correlated (Spearman's correlation) with SC and GS, respectively. We tested whether genetic similarity and structural covariance were higher within resting-state functional networks[34] or cytoarchitectonic classes by comparing the average genetic similarity (resp. structural covariance) of regions within networks (resp. classes) to regions between networks (resp. classes). We used spin-tests (1000 permutations) to test the significance of the co-location of two cortical maps while accounting for spatial-correlation between brain regions[96,97]. To control for multiple testing, we performed FDR-correction across all networks (resp. classes).

## Cophenetic correlations.

Cophenetic correlation coefficients were used to quantify the similarity of GS and SC network community structures within and between MRI metrics. To do this, Euclidean distance matrices were calculated using the `dist()` function, and `hclust(method = "ward)` was used to generate dendrograms for each phenotype. Cophenetic correlation coefficients were estimated using the `cor_cophenetic()` function from the dendextend R package[98]. To test for statistical significance, we generated a null distribution of cophenetic correlation coefficients by permuting the labels of one dendrogram 1000 times while keeping the dendrogram topologies constant.

## Hubness, modular decomposition, intra- and inter-modular degree.

Hubness for each brain region was defined by the weighted degree centrality, i.e., the sum of edge weights connecting each node to the rest of the network[20]. To ensure that regional degree was not driven by measurement reliability, we used follow-up scans from a subset of the UK Biobank cohort ($N = 1280$) to investigate the relationship between test-retest reliability of SA and CT and weighted degree of regional nodes in the corresponding structural covariance networks. We found no significant correlation between measurement reliability and hubness indexed by regional degree for either of these metrics (see SI Results 3.4 for details). To further characterise the network topology of structural covariance and genetic similarity matrices, we created weighted, undirected network graphs using `igraph` in R[99]. Modules were identified using the `cluster_louvain()` function, which implements a multi-level modularity optimisation algorithm for finding community structure[100]. For each node, intra-modular degree was computed as the mean weighted degree over all nodes in the same module. Inter-modular degree was estimated as the mean weighted degree over all nodes outside the module[16].

## Pleiotropic genes mediating covariation between schizophrenia and regional MRI metrics

**Partial least squares regression: PLS1 weights.** For each MRI metric, we extracted the first PLS (PLS1) component using the `plsdepot` package in R[101]. The significance of PLS1 was estimated by comparing the empirical variance explained by each component to a null distribution, i.e., the distribution of variance explained when permuting Y 1000 times. Additionally, we tested whether PLS weights were spatially enriched in known cortical atlases of network organisation (Yeo networks[34]) or cytoarchtectonic classes (Mesulam classes), and in identified modules by spin permutations. Spatial permutation testing (spin-tests) were used to control for spatial autocorrelation in PLS maps[96,97]. Results were FDR-corrected for each MRI metric and cortical atlas.

**Partial least squares regression: *T* and *U* scores.** To identify which genes most strongly impact the relationship between regional brain structure and schizophrenia, we investigated $T$ scores and $U$ scores, quantifying the contribution of each gene to the covariance between brain structure and schizophrenia. More formally, $T$ is defined as the product of $Xw_X$ and $U$ the product of $Yw_Y$. PLS maximises the covariance of $X$ and $Y$ with:

$$\max_{w_X w_Y} cov(Xw_X, Yw_Y) = corr(Xw_X, Yw_Y)\sqrt{var(Xw_X)}\sqrt{var(Yw_Y)} \quad (1)$$

Thus, covariance is a product of three terms: correlation (between $T$ and $U$ scores) and two standard deviations (of $T$ and $U$ scores). The correlation term informs about the true association between $T$ and $U$ (the standard deviations reflect variances within $X$ and $Y$), and given that correlation does not have a directionality (i.e., $T$ and $U$ are equally important), both $T$ and $U$ scores should be considered to select genes[32].

We used a leave-one-out (LOO) strategy, systematically re-estimating $R(T, U)$ after the exclusion of each gene in turn from the $X$ and $Y$ variables used for PLS estimation of $T$ and $U$ scores. This procedure resulted in 18,640 LOO estimates of the genetic relationship between brain MRI and schizophrenia phenotypes, each of them estimated after leaving out a single gene, $R_{LOO}(T, U)$. The influence of each gene on the whole genome relationship was then defined by

$$\Delta(R(T,U)) = R(T,U) - R_{LOO}(T,U) \quad (2)$$

i.e., genes that make the greatest individual contribution to the genetic relationship will have the largest positive values of $\Delta(R(T, U))$[32].

**Constrained genes, cell-type enrichment, gene ontology.** All enrichment analyses were conducted on the top 1% of genes, and repeated on the top 3% of genes, with the highest $\Delta(R(T, U))$ values; see SI 5.1 for details. We investigated enrichment in mutation-intolerant genes (pLOEUEF ≤ 0.37) identified by ref. 51 using logistic regression after accounting for log transformed gene length as a covariate. Since MRI-associated genes were highly expressed during mid-gestation, we used single cell RNA sequencing data from the mid-gestation period[102]. Specifically, expression values per cell were log-transformed and normalised. Mean cell type specific expression was divided by the average expression of genes in all cell types to calculate relative cell type expression. The average centred expression values of genes associated with each MRI modality were calculated for each cell type, and we performed linear regression analyses

controlled for log transformed gene length to assess significance[2,28]. Results were FDR corrected across cell types. We used the R package gProfiler2 for conducting GO enrichment analysis and focused on biological processes[94].

**Positional enrichment and local genetic correlation analysis of most pleiotropic genes.** Positional enrichments were performed using FUMA (functional mapping and annotation of genome-wide association studies). FUMA applies hypergeometric tests to investigate whether genes of interest are overrepresented in predefined chromosomal positions (MSigDB C1) while accounting for multiple comparison correction[38]. We used LAVA (Local Analysis of [co]Variant Annotation) to estimate the local SNP heritability and the genetic correlation between the schizophrenia GWAS and MRI GWAS for the brain regions with the highest PLS1 weights, i.e., the regions showing the highest genetic covariation with schizophrenia. To account for potential sample overlap, we estimated the intercepts from bivariate LD Score Regression as suggested by LAVA. Since the GWAS summary statistics were from European samples, we used the European panel of phase 3 of 1000 Genome as an LD reference. LAVA splits the genome into 2495 non-overlapping and broadly LD independent loci[39]. Since our primary goal was to identify sub-regions within the positionally enriched regions of the genome identified by FUMA, we restricted our analysis to loci that were within the positionally-enriched genomic regions leading to 71, 66 and 54 loci of interest for SA, CT, and NDI, respectively. For each pair of MRI plus schizophrenia phenotypes, genetic correlation analysis was only performed for loci in which both phenotypes had significant SNP heritability at $P \leq 0.05/71$ for SA, $P \leq 0.05/66$ for CT, $P \leq 0.05/54$ for NDI, resulting in 20, 14 and six tests for SA, CT and NDI, respectively. To identify significant genetic correlations, we used Bonferroni-corrected probabilities of type 1 error $P \leq 0.05/20$ for SA, $P \leq 0.05/14$ for CT, and $P \leq 0.05/6$ for NDI.

### Mendelian randomization
Genetic instruments were chosen at a $P$ threshold of $5 \times 10^{-8}$ and clumped with a distance of 10,000 kilobases (kb) and LD r-squared threshold (LD $r^2$) of 0.001. These SNPs were then identified in the outcome GWAS data, and SNP-level effects of exposure and outcome data were harmonised to match the effect alleles. To fit the MR models, we used inverse variance-weighted Mendelian randomization (IVW), implemented in the "twosampleMR" package v0.5.6[103], as the main method to estimate causal effects[103,104]. We also conducted a wide range of sensitivity analyses including weighted median (WM), MR Egger[104], Cochran's Q value[105], MR Presso[106], Steiger filtering[107] and we generated four types of plots for visual inspection (see SI Methods 7 for details).

### Reporting summary
Further information on research design is available in the Nature Portfolio Reporting Summary linked to this article.

### Data availability
Data used in this project are available as outlined below. Summary statistics of cortical MRI phenotypes are available for download from https://portal.ide-cam.org.uk/overview/483. Summary statistics for schizophrenia, bipolar disorder and Alzheimer's disease can be accessed from the Psychiatric Genomics Consortium https://pgc.unc.edu/for-researchers/download-results/. Height GWAS results are available at https://www.nature.com/articles/s41586-022-05275-y. MRI Data from the UKB can be applied for and accessed by approved researchers https://www.ukbiobank.ac.uk/. Spatiotemporal gene expression data can be accessed from PsychENCODE under http://development.psychencode.org/files/processed_data/RNA-seq/mRNA-seq_hg38.gencode21.wholeGene.geneComposite.STAR.nochrM.gene.count.txt. Cell type specific expression data can

be downloaded from http://solo.bmap.ucla.edu/shiny/webapp/. Information on constrained genes can be accessed from gnomAD https://gnomad.broadinstitute.org/downloads#v2-constraint. Gene sets for positional enrichments (MSigDB C1, version MSigDB 2023.2.Hs) can be found at https://www.gsea-msigdb.org/gsea/msigdb/human/collection_details.jsp#C1. Source data are provided with this paper.

### Code availability
Codes used are available at https://github.com/evastauffer/schizophrenia-and-brain-structure.

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

## Acknowledgements
E.-M.S. is supported by a PhD studentship awarded by the Friends of Peterhouse. This research was co-funded by the National Institute of Health Research (NIHR) Cambridge Biomedical Research Centre and a Marmaduke Sheild grant to R.A.I.B. and V.W. E.T.B. is an NIHR Senior Investigator. R.A.I.B. is supported by a British Academy Post-Doctoral fellowship and the Autism Research Trust. The views expressed are those of the author(s) and not necessarily those of the NHS, the NIHR or the Department of Health and Social Care. This research was possible due to an application to the UK Biobank (project 20904). We thank Dr Agoston Mihalik for his advice on partial least squares regression analysis.

## Author contributions
E.-M.S., R.A.I.B., V.W. and E.T.B. designed research. L.D. advised on PLS analysis. H.W. advised on research design. E.-M.S. analyzed data. E.-M.S. and E.T.B. wrote the paper.

## Competing interests
E.T.B. has consulted for GlaxoSmithKline, SR One, Boehringer Ingelheim, Sosei Heptares, and Monument Therapeutics. R.A.I.B. and E.T.B. are directors of and hold stock in CentileBio. All other authors declare no conflicts of interest.
