## [Peer Review File · Nature Communications]

The genetic relationships between brain structure and schizophreniaREVIEWER COMMENTS

Reviewer #1 (Remarks to the Author):

Stauffer et al performed an interesting study of genetic overlap between brain structure and schizophrenia, applying a range of methods, including MAGMA and PSL, analyzing large GWAS data from UK Biobank and schizophrenia GWAS.

Interesting findings, and support the recent evidence of genetic overlap, but the authors should clarify what is novel?

The major issue with the study is the approach to first annotate genes and then look for overlap at genetic level. They do SNP-to-gene mapping for each GWAS using H-MAGMA. This mapping strategy is inaccurate, and they should apply more fine mapping or other methods to ensure they have a reliable set of credible genes. As the whole approach builds on the gene mapping strategy, they should put more efforts into the functional annotation – using different methodological approaches, to ensure robust findings. It would be of interest to compare the results with analyses of overlap at the SNP level.

How are the genes identified for schizophrenia overlapping with the annotation analyses in the most recent PGC paper, Trubetsky et al where they obtain the GWAS data from? Here they found 587 genes, while the original publication reported 120 genes. It is strange that they claim discovery of schizophrenia genes not identified in the primary paper. Similar with brain structure genes. They should do the analyses with the credible genes from the Trubetsky et al paper.

They claim that many of these genes were uniquely associated with one of the three MRI metrics: SA, CT and NDI. It is unclear how they can prove if a specific gene is not associated with any brain measure. This seems to be related to statistical power, and the PLS findings are not helpful in clarifying the picture, as it is not reporting specific genes.

There are interesting overlaps and potential molecular mechanisms revealed, but the specificity of the findings is unclear – how much overlap is driven by non-specific associations? What is the overlap with non-schizophrenia phenotypes, like height? They need to compare with a control phenotype.

The Hub Node terminology should be better explained. This seems to be less relevant for biological mechanisms as it is building on statistical associations?

The functional characterization of the top genes with largest positive $\Delta(R(T,U))$ for each MRI metric seems ad hoc, as it was defined by top 1% (or 3%). What is the rationale for these thresholds?

Reviewer #2 (Remarks to the Author):

The authors provide an interesting and thorough study of shared genetic contributions to schizophrenia and brain structure, using data from large-scale studies together with state-of-the-art analyses. The results are consistent with what we know from previous studies of the genetics of schizophrenia and brain structure separately. This study will be a valuable contribution to the literature.

The introduction is framed around a causal mediation concept genome>brain>schizophrenia, i.e. brain structure may mediate effects of the genome on schizophrenia. I agree that this seems plausible for some genetic effects, but can the authors also consider other possibilities? For example, some genetic effects may be mediated through subtle changes to molecular or cellular physiology that lead to psychiatric symptoms, which then feed back onto brain structure secondarily, e.g. by affecting

medication behavior that affects brain structure, or changing dietary or lifestyle behavior that affects brain structure. I think the complexity of possible causal pathways would better be acknowledged up front in the manuscript. Finding genetic correlation between brain structure and schizophrenia does not disentangle these pathways.

With the structural covariance approach, I think it would be difficult for a region to achieve 'hub' status if its measurement was relatively noisy. The UK Biobank includes repeat scans so that repeatability across regions could be examined. This would help to understand whether repeatability is an issue when identifying hubs.

From the Methods: 'We used hypergeometric testing implemented in the R package GeneOverlap [72] to test for significant overlap between schizophrenia-associated genes and MRI metric-associated genes and performed permutation testing (10,000 permutations) to test whether this overlap is non-random'. Does this approach account for non-independence of gene-based associations caused by linkage disequilibrium? Given that the MAPT locus on chromosome 17 was implicated in this study, it seems possible that long-range linkage disequilibrium might have biased these results. In other words, a set of genes at this locus shows association with schizophrenia and also brain structure, but this could be only a single genetic signal, whereas I think the genes are entered into hypergeometric testing as multiple independent observations. Non-independence due to linkage disequilibrium may also have affected the PLS analysis and other aspects of the study.

From the Results: 'regions with high structural covariance had highly similar genetic profiles ... structural covariance and genetic similarity were greatest between regional nodes separated by the shortest geodesic distances'. Did the authors control for geodesic distance when assessing the relation between structural covariance and genetic similarity? It would be worth at least repeating the analysis with this control, as a sensitivity analysis. If geodesic distance accounts for the entire relation, then the finding seems less interesting/relevant.

The genetic results point most strongly to fetal brain development. How do the authors see this fitting with the typical age of onset for schizophrenia?

In the discussion it can be acknowledged that the analysis was limited to common SNPs, but rare genetic mutations also play a role in schizophrenia.

Point-by-point response

We are grateful to the editor and the two reviewers for providing us with valuable feedback on our manuscript. Their insightful comments and constructive criticism have contributed to improving our study. In this revision, we have carefully considered each point raised and have incorporated relevant amendments into the manuscript. Changes and additional analyses to the revised manuscript are highlighted in the response.

Reviewer #1 (Remarks to the Author):

Stauffer et al preformed an interesting study of genetic overlap ebetween brain structure and schizophrenia, applying a range of methods, including MAGMA and PSL, analyzing large GWAS data from UK Biboank and schizophrenia GWAS....with interesting findings, [that] support the recent evidence of genetic overlap,

We express our appreciation for the reviewer's time and effort in reviewing our manuscript. We are pleased that the findings generated interest.

1/1 But the authors should clarify what is novel?

We thank the reviewer for this question. Our study provides novel findings on several fronts.

- I. First, while it has long been hypothesised that schizophrenia and cortical brain phenotypes are genetically related, there is limited evidence to prove this hypothesis. Using three complementary approaches (overlap of significant genes, partial least squares and local genetic correlations), we report robust genetic overlap between the genetic risk for schizophrenia and cortical brain structure.
- II. Second, we show that the genetic relationship between schizophrenia and cortical brain structure converged positionally on three genomic regions, on chromosome 17q21, chromosome 3p21 and chromosome 11p11. While some of these regions were previously linked to schizophrenia, or a subset of brain MRI phenotypes, this is the first study to clearly implicate these regions by analysing the genetics of schizophrenia and cortical MRI phenotypes simultaneously and comprehensively.
- III. Third, we compare pleiotropic associations of regional brain phenotypes with schizophrenia to comparable results on pleiotropic association with bipolar disorder and Alzheimer's disease, highlighting the relative strength and largely disorder-specific pleiotropic association of schizophrenia with brain structure.
- IV. Fourth, we provide a biological basis for structural covariance networks observed in imaging data. Our findings suggest that cortical areas which are more similar and thus show a higher degree of connectivity have likely followed similar neurodevelopmental trajectories programmed by shared genetic mechanisms.
- V. Fifth, we show that normative network hubs are strongly associated with genes pleiotropically associated with schizophrenia, which is consistent with prior studies indicating that hub regions typically have the most atypical structure in case-control studies of schizophrenia.

- VI. Sixth, by Mendelian randomization analysis, we provide preliminary evidence for a causal relationship of a genetically determined brain structural phenotype (surface area of a region of posterior cingulate cortex) to the clinical outcome of schizophrenia.

In an effort to clarify these multiple novel findings, we have completely redrafted both the Abstract and Discussion.

1/2 The major issue with the study is the approach to first annotate genes and then look for overlap at genetic level. They do SNP-to-gene mapping for each GWAS using H-MAGMA. This mapping strategy is inaccurate, and they should apply more fine mapping or other methods to ensure they have a reliable set of credible genes. As the whole approach build on the gene mapping strategy, they should put more efforts into the functional annotation – using different methodological approaches, to ensure robust findings. It would be of interest to compare the results with analyses of overlap at the SNP level.

We thank the reviewer for their comments.

Whilst we appreciate the reviewer's suggestion of fine-mapping, we do not think fine-mapping is the right method for the questions at hand for three reasons. First, fine-mapping only focuses on genome-wide significant loci. For many of the regional GWAS, there were fewer than five genome-wide significant loci. Of the identified loci, only a subset will be fine-mapped to identify potential causal variants with a reasonably high posterior inclusion probability, and for a subset of these we will be able to link to genes using multiple methods. As such, fine-mapping will only identify a fraction of the genetic signal. Second, post fine-mapping there are numerous techniques that can be used to link the prioritised causal variant to genes including chromatin conformation capture, which is used in H-MAGMA. As such, it quickly becomes computationally infeasible to finemap significant loci for 180*3 GWAS, and then link them to genes using multiple different methods. Third, different fine-mapping tools produce different fine-mapping results, making it challenging to run downstream analyses.

However, we consider it important to ensure that our findings are robust to identification of genes using multiple different methodological approaches. Therefore, we have now performed several additional analyses to ensure that our results are robust. Specifically, we investigated the overlap between schizophrenia and brain structure using previously published lists of prioritized (fine-mapped) genes. These analyses are described in more detail in our response to the following reviewer comment **(1/3)**.

We also agree that it is valuable to test whether the genomic regions that mediate the covariation between brain structure and schizophrenia (including chromosome 17q21 and chromosome 3p21) are also identified using analyses that are performed on SNP level.

As outlined in the main manuscript, we performed local genetic correlations using LAVA. This method is based on SNP level data and tests for local heritability in both brain structure and schizophrenia traits, and tests for genetic correlations within non-overlapping and broadly LD independent loci (Werme, van der Sluis, Posthuma, & De Leeuw, 2021). Using LAVA, we identified significant local genetic correlations between surface area and schizophrenia within

the same regions that we identified using gene-level data including a subregion of chromosome 3p21 (3p21.2 - 3p21.1, start position: 51953969 - stop position: 54074844). This chromosomal region harbours genes that we reported as overlapping between significantly associated schizophrenia genes and significantly associated MRI genes including *NISCH* and *NEK4*. This chromosomal region and specifically those genes were also identified in the PLS analysis. Chromosome 17q21 was also identified using SNP-level data and LAVA. Specifically, we identified a significant local genetic correlation between SA and schizophrenia on chromosome 17q21.31 (start position: 43460501, stop position: 44865832). This genomic region harbours genes that were identified as overlapping using significantly associated schizophrenia genes and significantly associated MRI genes including *CRHR1* and *KANSL1*. Additionally, chromosome 17q21 was identified within the PLS analysis. Thus, using two different approaches, one based on gene-level data and one based on SNP-level data, we identified the same genomic regions that overlap between schizophrenia and MRI phenotypes.

Changes to Results section of main text

Since the loci of positional enrichment identified using FUMA were relatively long ($\geq 5\text{ Mb}$, $\leq 25\text{ Mb}$), and to ensure that positional enrichments are robust to methodological choices, we also estimated local genetic correlations at these positions to identify shorter genomic segments ($\sim 1\text{ Mb}$) mediating genetic covariation between MRI metrics and schizophrenia using LAVA (Local Analysis of [co]Variant Association) [35]. Local genetic correlations were based on summary statistics and thus represent a SNP-level approach to quantify the genetic overlap between schizophrenia and brain structure (Methods). Within four of the positionally enriched loci identified by FUMA, we were able to resolve the pleiotropic association to smaller subregions ($\geq 1\text{ Mb}$, $\leq 2\text{ Mb}$), e.g., chromosomal loci 14q32.2 - 14q32.31 and 14q32.13 for CT, and 3p21.2 - 3p21.1, 2q33.1 and 17q21.31 for SA (Fig.5 B). We did not find any significant local genetic correlations for NDI (Table S16).

1/3 How are the genes identified for schizophrenia overlapping with the annotation analyses in the most recent PGC paper, Trubetsky et al where they obtain the GWAS data from? Here they found 587 genes, while the original publication reported 120 genes. It is strange that they claim discovery of schizophrenia genes not identified in the primary paper. Similar with brain structure genes. They should do the analyses with the credible genes from the Trubetsky et al paper.

We thank the reviewer for their interesting question and the suggestion to investigate the gene-level overlap between MRI phenotypes and schizophrenia using previously published lists of prioritized or fine-mapped genes. Genes can be identified using multiple different approaches with varying levels of inclusivity. For example, Trubetsky et al. (2022) published a broad fine-map set which included 628 genes (435 protein-coding), as well as a prioritized list of 120 genes (106 protein-coding), associated with risk of schizophrenia. We performed several sensitivity analyses to ensure that our findings are robust. First, we investigated the overlap between the schizophrenia-associated genes we identified and the 106 protein-coding genes, and the 435 protein-coding genes from the broad fine-map gene set, reported by Trubetsky (2022). Second, we investigated the gene level overlap between these 106 schizophrenia-associated protein-coding genes and the list of prioritised genes published by

Warrier et al. (2022). Third, since the number of genes associated with MRI phenotypes was low, we additionally performed enrichment analysis to test whether MRI-associated genes were enriched for the 106 schizophrenia-associated genes.

Changes to Results section of main text

Whilst H-MAGMA aggregates genome-wide SNP level data into gene-level data, an alternative method is to identify genes by fine-mapping significant loci, and linking these to genes, as has been done previously for both schizophrenia [20] and global MRI phenotypes [3]. As a sensitivity analysis, we also investigated if genes prioritised from fine-mapping analyses of schizophrenia (number of genes, $N_G = 106$) [20] are enriched for genes identified from fine-mapping of global MRI metrics (SA, $N_G = 16$; CT, $N_G = 12$; NDI, $N_G = 2$) [3]. The gene effects we identified for SA and CT were enriched for the list of prioritized schizophrenia risk genes ($N_G = 106$) [20], and we replicated the genetic intersection between schizophrenia and both SA and CT metrics that was located on chromosome 17q21.31 using the previously published fine-mapped gene lists [3,20] (see SI Results 2).

Changes to Supplementary Information

2. Genetic intersection of brain structure and schizophrenia was robust to gene inclusion criteria

Genes can be identified using multiple different approaches with varying levels of inclusivity. For example, Trubetsky et al. published a broad fine-map set which included 628 (435 protein-coding) genes as well as a prioritized list of 120 genes (106 protein-coding). To ensure that the identified genes which overlapped between schizophrenia and brain structure were robust to the method used for gene identification, we performed three sensitivity analyses. First, we investigated the overlap between the 587 schizophrenia-associated genes we identified and the two lists of schizophrenia-associated genes reported by Trubetsky et al., [9] (106 protein-coding genes and a more inclusive list of 435 genes). Second, we investigated the gene-level overlap between the 106 schizophrenia-associated genes and the list of brain MRI-associated genes published by Warrier et al. [3]. Third, because the number of genes significantly associated with MRI phenotypes was low, due to the relatively small sample size of MRI GWAS, we additionally performed enrichment analysis to test whether MRI-associated genes were enriched for the 106 schizophrenia-associated genes.

First, we found that out of the 106 protein-coding genes reported by Trubetsky et al. [9], 42 genes (40%) were also included in our list of genes associated with schizophrenia. Additionally, out of the 435 protein-coding genes listed in the more inclusive set of schizophrenia-associated genes [9], 198 (46%) were also identified by our analysis. Importantly, some of the genes robustly associated with schizophrenia across these various lists were genes that showed strong effects on the genetic covariation between schizophrenia and all MRI phenotypes, e.g., CRHR1, KANSL1 and MAPT on chromosome 17q21 and ATG13 on chromosome 11p11. Second, the prioritised gene lists for MRI phenotypes reported by Warrier et al. [3] included 16 genes for SA, 12 for CT and two for NDI, all of which were also identified using our gene-level analysis. From these two publications [3,9], we identified four genes that were robustly associated with schizophrenia and with SA, three genes with CT, and three with NDI. These consistently overlapping genes included BNIP3L on chromosome 8p21 and CRHR1, MAPT and KANSL1 on

chromosome 17q21. Third, we found that gene effects for surface area and cortical thickness were enriched for 106 fine-mapped genes for schizophrenia (surface area $Z = 2.6$, $P < 0.02$; cortical thickness $Z = 2.6$, $P < 0.01$). Gene-level effects for neurite density index were not significantly enriched for schizophrenia-related genes ($Z = 0.8$, $P = 0.4$). Taken together, these additional analyses support our findings of overlap between schizophrenia- and MRI-associated genes. Specifically, using more stringent gene lists for both schizophrenia and MRI phenotypes, we replicated the overlap of schizophrenia- and brain-related genes on chromosome 17q21.

1/4 They claim that many of these genes were uniquely associated with one of the three MRI metrics: SA, CT and NDI. It is unclear how they can prove if a specific gene is not associated with any brain measure. This seems to be related to statistical power, and the PLS findings are not helpful in clarifying the picture, as it is not reporting specific genes.

We thank the reviewer for their comment and agree that we cannot prove that a specific gene is *not* associated with any brain measure. We have clarified our reporting of the relevant results accordingly and we have added a brief discussion of this point in the main paper:

Changes to Results section of main text

Most of these genes were associated specifically with one of the three MRI metrics investigated: 246 out of 318 genes (78%) associated with SA were associated only with SA; 95 out of 157 genes (61%) were associated only with CT; and 43 out of 86 genes (63%) were associated only with NDI. This parallels the minimal genetic correlations across these three MRI metrics [2]. However, 27 genes were significantly associated with all MRI metrics, including 16 genes within the 17q21.31 region. The remaining (11) genes associated with all 3 MRI metrics were located on chromosome 8p23 (7 genes), chromosome 6q25 (3 genes), and chromosome 1p33 (1 gene) (Fig.1B, Table S2).

Changes to Discussion section of main text

Most genes associated with any brain phenotype were only associated with one phenotype, indicating some specificity of genetic effects for distinct MRI metrics. This must be caveated by the limited number of MRI metrics considered, and the relatively small sample sizes currently available for marginally powered GWAS of any MRI phenotypes. However, genetic associations specific to different MRI metrics are compatible with previous studies demonstrating that CT, SA and NDI are genetically relatively distinct [1,2].

1/5 There are interesting overlaps and potential molecular mechanisms revealed, but the specificity of the findings is unclear – how much overlap is driven by non-specific associations? What is the overlap with non-schizophrenia phenotypes, like height? They need to compare with a control phenotype.

We thank the reviewer for their comment, and we agree that it is valuable to compare our reported findings with a non-schizophrenia, positive control phenotype. To assess how much of the overlap between brain structure and genetic risk for schizophrenia is shared with other brain disorders, we repeated a large number of the principal analyses using additional new

GWAS data relevant to bipolar disorder and Alzheimer's disease. We expected that genetic risks for schizophrenia and bipolar disorder would share some degree of overlap with the genetics of brain structure, reflecting their clinical similarity as psychotic disorders of developmental origin; whereas the genetic risks for Alzheimer's disease were expected to overlap with the genetics of brain structure more distinctively, reflecting its clinical differentiation as a cognitive disorder of neurodegenerative aetiology.

Changes to Abstract

Parallel analyses of GWAS on bipolar disorder and Alzheimer's disease showed that pleiotropic association with MRI metrics was stronger for schizophrenia compared to other disorders.

Changes to Introduction section of main text

...These results prompted us to address two more (secondary) questions: (iv) how does schizophrenia compare to other brain disorders in terms of its shared genetic risk with brain structure?...

Changes to Results section of main text

Clinical diagnostic specificity of genetic covariation between schizophrenia and brain structure

We repeated many of the principal analyses of pleiotropic association with schizophrenia, using identical methods and models applied to independent data on two additional disorders: large-scale GWAS for bipolar disorder (BIP) (N = 41,917 cases and N = 371,549 controls) [41] and Alzheimer's disease (AD) (N = 398,058, excluding UK Biobank and 23andMe participants) [42].

We identified genes significantly associated with BIP (or AD) and investigated their intersection with MRI-associated genes. Out of 136 BIP-associated genes, only the intersection with 15 genes also associated with SA was significant, and largely comprised genes located at chromosome 3p21. Out of 77 AD-associated genes, there were significant intersections with genes also associated with SA (15), CT (12), and NDI (11); and several genes associated with both AD and surface area were located at chromosome 17q21. We also used PLS regression, as previously for analysis of whole-genome covariation with schizophrenia (Fig.2), for comparable analysis of regional brain phenotypes that were pleiotropically associated with risk of BIP or AD; see Fig.6 and SI Results 6 for details.

Figure 6. Specificity of pleiotropic associations between clinical disorders and MRI metrics. (A) Proportion of variance in the genetically predicted risk for each disorder (y-axis) explained by the genetic effects on MRI metrics (x-axis; SA = surface area, CT = cortical thickness, NDI = neurite density index) based on the first PLS component, PLS1. The proportion of disorder-related variance explained was greater for schizophrenia (SCZ: SA = 5.9% , CT = 5.5%, NDI = 3%), than for bipolar disorder (BIP; SA = 2.7%, CT = 3.6%, NDI = 1.6%), or Alzheimer’s disease (AD; SA = 1.5%, CT = 1.2%, NDI = 0.8%). (B) Cortical surface maps of PLS1 regional scores for SCZ, BIP, and AD. Higher positive weights (shades of yellow) indicate stronger genetic covariation with each disorder; regions with zero weight are shown in white. Mean absolute weights were lower for BIP (SA \bar{w} = 12.3, CT \bar{w} = 9.45, NDI \bar{w} = 8), and for AD (SA \bar{w} = 9, CT \bar{w} = 5.2, NDI \bar{w} = 5.4), than for schizophrenia (SA \bar{w} = 18.29, CT \bar{w} = 11.89, NDI \bar{w} = 11.37). Fewer brain regions had significant PLS1 scores for BIP (NDI = 175) and AD (SA = 79, CT = 166, NDI = 170) than for schizophrenia (SA, CT = 180, NDI = 179). (C) Spearman’s correlations (y-axis) between T and U scores for schizophrenia, bipolar disorder, and Alzheimer’s disease. The strength of pleiotropic association indexed by ρ was greater for schizophrenia (SA ρ = 0.24, CT ρ = 0.23, NDI ρ = 0.17), than for BIP (SA ρ = 0.17, CT ρ = 0.19, NDI ρ = 0.13) or AD (SA ρ = 0.12, CT ρ = 0.11, NDI ρ = 0.09). (D) Venn diagrams showing the intersection of the top 1% of genes with the highest $\Delta(R(T,U))$ scores for schizophrenia and bipolar disorder (top row) and for schizophrenia and Alzheimer’s disease (bottom row). The number of overlapping genes was less than 50% for bipolar and schizophrenia and further decreased for Alzheimer’s disease and schizophrenia.

Changes to Supplementary Information, Results

6. Specificity of schizophrenia’s intersection with MRI-associated genes

We compared the genes we identified as significant for bipolar disorder (BIP) and Alzheimer’s disease (AD) to previously published fine-mapped or prioritized gene lists from prior GWAS to ensure that our lists overlap with fine-mapped genes. As outlined in the main manuscript, we found 136 genes were significantly associated with bipolar disorder after correction for

multiple comparisons. 19 of them were also reported in the original paper [21] including prioritized genes such as *GNL3*, *TMEM258* and *STK4*. For Alzheimer's disease, we found 76 genes were significant after multiple comparisons correction. 55 of those genes overlapped with the 989 genes that were identified in the original publication [22] using position and expression quantitative trait loci, including high confidence genes such as *CD33* and *MADD*.

As outlined in the main manuscript the first PLS component for each MRI metric identified a small but significant proportion of its genetically determined variation that covaried with genetic risks for bipolar disorder (2.7% for SA, 3.6% for CT and 1.6% for NDI). However, the variance explained for BIP was about 50% less than the variance explained for schizophrenia by the same MRI metrics; and the variance explained for AD was about 75% less than for schizophrenia (1.5% for SA, 1.2% for CT and 0.8% for NDI)(Fig.6A).

The strength of pleiotropic association with schizophrenia and MRI phenotypes, across all 18,640 genes, was also greater for schizophrenia than for BIP or AD. Specifically, the correlation between T and U scores decreased from schizophrenia, to bipolar disorder to Alzheimer's disease as shown in Figure 6 C. Finally, we visualised the intersection of the top 1% of genes with the highest $\Delta(R(T,U))$ values between schizophrenia and bipolar disorder or Alzheimer's disease respectively. The number of overlapping genes was less than 50% for bipolar and schizophrenia and further decreased for Alzheimer's disease and schizophrenia (Fig.6D). In summary, these results suggest that the genetic covariation between brain structure and schizophrenia is stronger than with bipolar disorder and Alzheimer's disease, and that more than 50% of the genes investigated in down-stream analysis were specific to schizophrenia.

Changes to Discussion section of main text

However, they raised two secondary questions, about specificity and causality. The question of clinical diagnostic specificity (iv) is whether there are similar genetic associations with both brain phenotypes and risk for other neuropsychiatric disorders. We repeated the analysis for pleiotropic associations with regional brain phenotypes, exactly as we had done for schizophrenia, using large GWAS datasets on bipolar disorder and Alzheimer's disease. We found that genetic covariation with brain structure was stronger for schizophrenia than for bipolar disorder or Alzheimer's disease, and pleiotropic genes were largely specific to each disorder, although there were also some genes and loci that were pleiotropically associated with more than one brain disorder. For example, the chromosome 17q21 locus was pleiotropically associated with cortical surface area and both schizophrenia and Alzheimer's disease (it has previously been linked to AD alone [58]); and the chromosome 3p21 locus was pleiotropically associated with surface area and both schizophrenia and bipolar disorder (it has previously been linked to BIP alone [71]). These are plausible but preliminary results and further studies, also integrating rare variants, will clearly be needed to survey the commonalities and differences between brain disorders in terms of their genetic relationships with brain structure.

1/6 The Hub Node terminology should be better explained. This seems to be less relevant for biological mechanisms as it is building on statistical associations?

We thank the reviewer for their comment and we have extended the explanation for hub nodes accordingly.

Changes to Introduction section of main text

*Recognising that the cortex is organised as a complex network [14], we also considered it important to investigate genetic associations with brain network phenotypes. For example, hubs or highly connected nodes have been demonstrated across a wide range of scales and species of nervous systems, including the neuronal network of *C. elegans* and axonal tract-tracing connectomes of mouse, rat and non-human primate brains [15,16,17,18]. In a structural covariance network derived from human MRI, hubs typically represent regions that covary strongly with multiple other cortical areas on some MRI metric, e.g., volume or thickness. This phenotypic covariance can be interpreted biologically as a proxy for axonal connectivity and/or shared neurodevelopmental trajectories between strongly covarying regions [19]; and twin studies have shown that human MRI network hubs are heritable [20].*

1/7 The functional characterization the top genes with largest positive $\Delta(R(T,U))$ for each MRI metric seems ad hoc, as it was defined by top 1% (or 3%). What is rationale for these thresholds?

Thank you for raising this question. As shown in the main manuscript and in Figure 5A, the PLS analysis revealed specific genes with the strongest pleiotropic effect on schizophrenia and brain structure. These genes are annotated in Figure 5A. In a first step, we discuss these genes in the discussion and contextualise our findings with previous studies implicating the same genes (e.g. *PLEKHM1* and *CRHR1*). However, since both schizophrenia and brain structural phenotypes are polygenic, we considered it important to go beyond the characterisation of a small number of genes and instead to characterise gene sets with large effects. Since a substantial number of genes had very small or no effects, which would bias down-stream analyses, we chose a thresholding approach. To do this we chose two thresholds (top 1% and top 3%) for gene selection to include genes with the largest effects $\Delta(R(T,U)) \geq 2$ or ≤ -2 (1%) and $\Delta(R(T,U)) \geq 1$ or ≤ -1 (2%) respectively. We performed the analyses on two thresholds to ensure that the findings are robust. We have added the following clarifications to the manuscript.

Changes to Methods section of main text

All enrichment analyses were performed on the top 1 % of genes with the highest $\Delta(R(T,U))$ values and repeated on the top 3% of genes with the highest $\Delta(R(T,U))$ values; see SI 5.1 for details.

Changes to Supplementary Information, Results

5.1 Thresholds for $\Delta(R(T,U))$ used to define gene sets for enrichment analysis

Since both schizophrenia and brain structural phenotypes are polygenic and likely influenced by genes that do not reach genome-wide significance, we considered it important to go beyond the characterisation of a small number of genes and additionally to characterise gene sets with large pleiotropic effects. However, as shown in Figure S7 A, many of the genes

showed weak effects which would bias downstream enrichment analysis. We therefore selected genes based on thresholds for $\Delta(R(T,U))$ chosen to avoid the inclusion of genes that have no or very low pleiotropic effects on brain MRI phenotypes and schizophrenia (or any other clinical disorder for which there is GWAS data). We thus included a larger proportion of genes with a relatively strong effect on the covariation between schizophrenia and brain structure (top 1%) to perform gene set enrichment analysis. This first threshold is conservative and only includes 185 genes with pleiotropic effects indexed by $\Delta(R(T,U)) \geq 2$ or ≤ -2 (Fig.S7 B). To ensure that the reported enrichment findings are robust across different inclusion thresholds, we widened the analysis to include the top 3% of genes. This threshold was chosen because it allows the inclusion of more genes (556) with effects indexed by $\Delta(R(T,U)) \geq 1$ or ≤ -1 (Fig.S7 C). As reported in the main paper, enrichment results based on the top 1% gene set were robustly replicated in the top 3% gene set.

Figure S7. Thresholds for pleiotropic gene sets for enrichment analyses. (A) Scatterplot of T scores (x-axis) versus U scores (y-axis) for each of 18,640 protein-coding genes, derived from their weights on the first PLS component. The number of genes is shown as a heatmap (count). (B) Scatterplot of T scores (x-axis) versus U scores (y axis) for top 1% of genes with highest $\Delta(R(T,U))$ scores (see Methods). (C) Same as in (B) but for top 3% of genes on $\Delta(R(T,U))$. Red lines are drawn at effect sizes of 1 and -1.

Reviewer #2 (Remarks to the Author):

The authors provide an interesting and thorough study of shared genetic contributions to schizophrenia and brain structure, using data from large-scale studies together with state-of-the-art analyses. The results are consistent with what we know from previous studies of the genetics of schizophrenia and brain structure separately. This study will be a valuable contribution to the literature.

We appreciate the reviewer's evaluation and constructive feedback.

2/1 The introduction is framed around a causal mediation concept genome>brain>schizophrenia, i.e. brain structure may mediate effects of the genome on schizophrenia. I agree that this seems plausible for some genetic effects, but can the authors also consider other possibilities? For example, some genetic effects may be mediated through subtle changes to molecular or cellular physiology that lead to psychiatric symptoms, which then feed back onto brain structure secondarily, e.g. by affecting medication behavior that affects brain structure, or changing dietary or lifestyle behavior that affects brain structure. I think the complexity of possible causal pathways would better be acknowledged up front in the manuscript. Finding genetic correlation between brain structure and schizophrenia does not disentangle these pathways.

We thank the reviewer for their comment and agree with the reviewer that there are likely causal mediation concepts that do not necessarily follow the outlined genome > brain > schizophrenia structure. We also agree that correlations between brain structure and schizophrenia do not inform about potential causal relationships. In addition to adapting the introduction, we performed Mendelian randomization (MR) analysis to investigate the causal relationship between MRI metrics and schizophrenia:

Changes to Introduction of main text

We reasoned that identification of such pleiotropically associated genes would be consistent with prior theories that genetic variants encode risk for schizophrenia by causal effects on intermediate endo-phenotypes of brain structure. We recognised that pleiotropic association per se does not resolve the question of causality, and that the theoretically privileged axis - gene > brain > schizophrenia - is not the only plausible causal pathway between these entities [9]. However, given recently available statistically well-powered GWAS of schizophrenia and brain structure, we reasoned that if we could not find any evidence for pleiotropic association, then the role of macro-scale brain structure in mediating schizophrenia risk must be more modest than previously anticipated [10, 11].

...These results prompted us to address two more (secondary) questions: (iv) how does schizophrenia compare to other brain disorders in terms of its shared genetic risk with brain structure? and (v) given this pleiotropy, is there a causal pathway for brain-mediated genetic risk of schizophrenia?

Changes to Results section of main text

Causal relationships between brain and schizophrenia phenotypes

We used two-sample Mendelian randomization (MR) analysis to test two directions of causal relationship between brain and schizophrenia phenotypes: (i) schizophrenia (exposure) causing brain changes (outcome); and (ii) brain changes (exposure) causing schizophrenia (outcome). We restricted Mendelian randomization analysis to a subset of regional MRI metrics that showed ≥ 5 genome-wide significant loci, to ensure reasonable statistical power. This condition was not satisfied for all cortical areas by any MRI metric: out of 180 regions, 48 had ≥ 5 gene-level associations with SA, but there were only 10 regional NDI phenotypes and 5 regional CT phenotypes which passed the criterion. After correcting for multiple comparisons with FDR 5%, we did not find any significant evidence for a causal effect of schizophrenia on the cortical thickness, surface area or NDI of this subset of brain regions.

However, there was evidence for a significant causal effect of genetically predicted brain structure on schizophrenia (Fig. S10). Specifically, SA of V4 and ProS cortical areas was predictive of risk for schizophrenia (inverse variance weighted method: V4, $\beta = 0.38$, $SE = 0.1$, $P = 0.02$; ProS, $\beta = 0.26$, $SE = 0.05$, $P = 0.0002$). For ProS (prostriate cortex), a region of posterior cingulate cortex, sensitivity analyses indicated that the effect of this exposure on the outcome of schizophrenia was robust and not attributable to horizontal pleiotropy. For V4, a region of ventral occipital cortex specialised for color vision, sensitivity analyses were less consistent and indicated potential horizontal pleiotropy. See Methods and SI Results 7 for details.

Changes to Methods section of main text

Genetic instruments were chosen at a P threshold of 5×10^{-8} and clumped with a distance of 10,000 kilobases (kb) and Linkage Disequilibrium r -squared threshold (LD r^2) of 0.001. These SNPs were then identified within the outcome GWAS, and SNP-level effects of exposure and outcome data were harmonised to match the effect alleles. To fit the MR models, we used inverse variance-weighted Mendelian randomization (IVW), implemented in the 'twosampleMR' package v0.5.6 [95], as the main method to estimate causal effects [95,96]. We also conducted a wide range of sensitivity analyses including weighted median (WM), MR Egger [96], Cochran's Q value [97], MR Presso [98], Steiger filtering [99] and we generated four types of plots for visual inspection (see SI Methods 7 for details).

Changes to Supplementary Information, Methods & Results

7 Mendelian Randomization

As summarised in the main paper, we used Mendelian randomization (MR) analysis to investigate the causal relationships between genetically coupled phenotypes, brain structure and schizophrenia, each of which could be regarded as both outcome and exposure. For the principal analysis, we used invariance weighted (IVW) estimators of MR model parameters. There was no evidence for schizophrenia exposure causing brain change outcomes; however, there was some evidence for genetically determined brain changes (exposure) causing schizophrenia (outcome). Specifically, there were significant causal effects based on the invariance-weight method in two brain regional phenotypes: ProS surface area and V4 SA.

IVW assumes that all SNPs are valid genetic instruments and that there is no horizontal pleiotropy and is thus not robust to horizontal pleiotropy [12]. Horizontal pleiotropy refers to the fact that a genetic variant (instrument) can be independently associated with multiple phenotypes. For example, a genetic variant can be associated with the outcome by a causal pathway through the exposure, and through an alternative causal pathway that does not include the exposure. Such horizontal pleiotropy contravenes one of the basic assumptions of MR analysis and can bias its results [13]. To assess the robustness of significant findings obtained using IVW and to investigate horizontal pleiotropy we conducted a series of sensitivity analysis.

We repeated our analysis using two robust MR methods that relax the assumption that there is no horizontal pleiotropy: the weighted median method (WM) and MR-Egger. WM provides consistent results even when 50% of the genetic instruments are invalid [12]. MR-Egger is a pleiotropy-robust method that allows for (some) directional pleiotropy, by including an intercept term in the IVW model. The slope from an MR-Egger regression represents the MR-Egger estimate of the causal effect. The intercept in an MR-Egger regression model is zero in the ideal case of no horizontal pleiotropy and significantly non-zero MR-Egger intercepts indicate substantial horizontal pleiotropy [12, 13, 14]. We additionally assessed heterogeneity of the genetic instruments using Cochran's Q value. If there is no horizontal pleiotropy, the MR estimates of causality for each individual SNP should be consistent and will only vary by chance. Thus, larger between-instrument heterogeneity, where effect estimates are more different than expected by chance, would indicate violation of the basic assumptions of MR analysis [15].

Two additional sensitivity analyses used the MR Presso global test, which detects the presence of horizontal pleiotropy [16], and Steiger filtering, which tests for the direction of effect [17]. We also generated four types of plots for visual inspection: (i) scatter plots showing the SNP effects on exposure versus SNP effects on the outcome. (ii) forest plots showing the variant-specific causal estimate for each individual genetic instrument (also known as Wald ratios) [18], combined with the overall estimates. (iii) leave-one-out plots showing the estimated causal effect of the exposure on the outcome after the exclusion of each genetic instrument, combined with the overall IVW; and (iv) funnel plots displaying the individual Wald ratio for each SNP versus its precision. Plots (i-iii) were used to detect genetic instruments that were potential outliers. Plot (iv) was used to assess unbalanced horizontal pleiotropy, which could bias the results of MR analysis [13, 19], and would be indicated by an asymmetric distribution of the variants around the estimate.

Sensitivity analyses using the weighted median method were also significant for both SA regions (V4 $\beta = 0.46$, $SE = 0.07$, $P \leq 0.0001$; ProS $\beta = 0.29$, $SE = 0.07$, $P = 0.0001$).

For ProS (prostriate cortex, an area of posterior cingulate cortex) the MR-Egger intercept ($P = 0.16$), the Q-test ($P = 0.52$) and the global MR Presso ($P = 0.49$) tests were not significant, suggesting no evidence for horizontal pleiotropy; and the Steiger test indicated correct causal direction ($P \leq 0.0001$). In addition, leave-one-out analyses did not indicate that the results were driven by any one genetic variant (Fig.S10).

For V4, an area of ventral occipital cortex specialised for colour vision, the sensitivity analyses were less consistent. The Egger intercept $P = 0.09$ did not reach significance but approached significance, implying that there might be pleiotropy present. Additionally, the Q-test was significant $P = 0.01$, thus indicating horizontal pleiotropy. However the global MR Presso test was not significant $P = 0.1$ and the Steiger test indicated correct direction of effect $P \leq 0.0001$. Taken together this findings should thus be interpreted with caution (Fig.S10).

Figure S10. Mendelian randomization plots for surface area of cortical areas, ProS and V4. (A,E) Scatter plots showing the SNP effect on exposure (x-axis) and on the outcome (y-axis). The regression lines represent the causal estimates based on the inverse variance weighted method (light blue), MR Egger (blue) and the weighted median method (green). (B,F) Forest plots showing the Wald ratios (i.e. variant-specific causal estimate, x-axis) of each individual genetic instrument, combined with the overall causal estimates for all three methods. (C,G) Leave-one-out plots showing the causal estimate (x-axis) after the exclusion of each genetic instrument, combined with the overall IVW estimate. (D,H) Funnel plots displaying the individual Wald ratio for each SNP (x-axis) against their precision (y-axis).

Changes to Discussion section of main text

The question of causality (v) arises because pleiotropy is a necessary (but not a sufficient) condition for the traditional causal model of biological pathogenesis: that genetic variation causes brain changes which in turn cause schizophrenia [8]. Certainly this model was not refuted by lack of evidence for pleiotropic association in this study. However, none of these results can resolve the causal relationship between the two genetically coupled phenotypes: do brain phenotypes cause schizophrenia or vice versa? Mendelian randomization provides a potentially powerful approach to address this question more directly and we used it to test both the standard causal pathway - gene > brain > schizophrenia - and the alternative causal pathway - gene > schizophrenia > brain (as previously reported for frontal cortex [9]). We

found no evidence for the alternative pathway and limited evidence for the standard pathway. Genetically predicted surface area of two cortical regions (ProS and V4) was predictive of schizophrenia, and the effect of ProS on schizophrenia was robust to sensitivity analyses. The posterior cingulate cortex has previously been associated with polygenic risk scores for schizophrenia [5, 71, 72] and case-control studies have reported abnormalities of macro- and micro-structural MRI metrics in this region [73, 74, 75, 76]. It seems plausible that genetically determined changes in surface area of the posterior cingulate cortex might cause increased risks of schizophrenia, as suggested by these results. However, the MR analyses that we were able to do were limited. Only a minority of regional brain phenotypes had sufficient, robustly significant genetic instruments for MR analyses. This reflects the relatively small size of currently available MRI GWAS datasets [1,2], e.g., compared to GWAS for schizophrenia [26] or educational attainment, which likely constrained our statistical power to detect multiple small-effect gene variants associated with many brain phenotypes [9,77] For more definitive future investigations of the fundamentally important question of causal relationships between brain structure and brain disorders it will be essential to build larger GWAS datasets for anatomically comprehensive and technically diverse MRI phenotypes.

2/2 With the structural covariance approach, I think it would be difficult for a region to achieve ‘hub’ status if its measurement was relatively noisy. The UK Biobank includes repeat scans so that repeatability across regions could be examined. This would help to understand whether repeatability is an issue when identifying hubs.

The reviewer is correct that structural covariance will tend to zero if the measurements at one or both nodes are randomly noisy. However, the datasets used for this analysis were carefully quality controlled to exclude noisy scans. For example, as outlined in **SI methods 1**, we excluded scans that were outliers both on global metrics as well as within each of the 180 brain regions. For the present analyses, we specifically focused on measures that are widely used in the neuroimaging community and have been well validated throughout the neuroimaging literature, including by the UK Biobank consortium providing the data (E. Haddad et al., 2023; Hedges et al., 2022; Knusmann et al., 2022). With regards to using follow-up data, the UK BioBank has indeed collected repeat scans to assess test-retest reliability and all the metrics we have analysed have been shown to have good-to-excellent reliability in these data (Duff et al 2021).

Changes to Supplementary Information, Methods:

In the present analyses we focused on MRI metrics that have been widely used and well-validated in the neuroimaging community, with multiple studies by the UK Biobank consortium providing the data [7, 8, 9], including results which have demonstrated that all 5 metrics have high levels of test-retest reliability in a repeatedly scanned subset of the UKB MRI cohort [10].

2/3 From the Methods: ‘We used hypergeometric testing implemented in the R package GeneOverlap [72] to test for significant overlap between schizophrenia-associated genes and MRI metric-associated genes and performed permutation testing (10,000 permutations) to test whether this overlap is non-random’. Does this approach account for

non-independence of gene-based associations caused by linkage disequilibrium? Given that the MAPT locus on chromosome 17 was implicated in this study, it seems possible that long-range linkage disequilibrium might have biased these results. In other words, a set of genes at this locus shows association with schizophrenia and also brain structure, but this could be only a single genetic signal, whereas I think the genes are entered into hypergeometric testing as multiple independent observations. Non-independence due to linkage disequilibrium may also have affected the PLS analysis and other aspects of the study.

We thank the reviewer for their important question. It is correct that the hypergeometric test for overlap and the PLS analyses do not account for linkage disequilibrium (LD). To ensure that our findings of genes pleiotropically associated with schizophrenia and brain structure were not biased by LD, we conducted additional gene set enrichment analysis using MAGMA, which accounts for LD between genomic regions. Specifically, we tested: (i) if the genetic effects of schizophrenia were enriched for sets of genes significantly associated with each MRI metric; and (ii) if the genetic effects of each MRI metric were enriched in a gene-set significantly associated with risk for schizophrenia. We found significant enrichments in both directions for all investigated MRI metrics.

Taken together with the results based on hypergeometric testing reported in the main manuscript (“Genes associated with schizophrenia and their intersection with MRI-associated genes”), these new results provide further confidence that the reported evidence for genetic overlap between schizophrenia and MRI metrics is not simply due to LD. As previously noted in response to comment **1/3**, we additionally investigated the overlap between schizophrenia risk genes and MRI-associated genes using lists of prioritized genes from previous publications (Trubetskoy et al., 2022; Warrier et al., 2022) and found converging results, particularly for genes located on chromosome 17q21.

Changes to Results section of main text

To ensure that pleiotropic association with schizophrenia and MRI metrics was not driven by linkage disequilibrium (LD) between genomic variants, we performed gene set enrichment analysis using MAGMA, which accounts for LD between genes (Methods) [30]. We found that the genetic effects of schizophrenia were enriched for genes significantly associated with each MRI metric (SA, $P < 0.0001$; CT, $P < 0.001$; NDI, $P < 0.01$); and, vice versa, that genes associated with MRI phenotypes were enriched for schizophrenia-related genes (SA, $P < 0.0001$; CT, $P < 0.001$; NDI, $P < 0.05$). These results provide confidence that the evidence for pleiotropic association is not simply driven by LD.

Changes to Methods section of main text

Additionally, we performed gene-set enrichment analysis using MAGMA, which accounts for linkage disequilibrium between genetic variants [30]. First, we tested whether the genetic effects of schizophrenia were enriched for gene-sets significantly associated with each MRI metric. Gene-sets for MRI metrics included all the genes that we identified as significant for each MRI metric (SA $N_G = 318$; CT $N_G = 157$; NDI $N_G = 86$). Second, we tested whether the genetic effects of each MRI metric were enriched in a gene-set significantly associated with risk for schizophrenia ($N_G = 587$).

2/4 From the Results: ‘regions with high structural covariance had highly similar genetic profiles ... structural covariance and genetic similarity were greatest between regional nodes separated by the shortest geodesic distances’. Did the authors control for geodesic distance when assessing the relation between structural covariance and genetic similarity? It would be worth at least repeating the analysis with this control, as a sensitivity analysis. If geodesic distance accounts for the entire relation, then the finding seems less interesting/relevant.

We thank the reviewer for this valuable remark. As suggested by the reviewer, we extended our analysis by using partial correlations to control for linear effects of geodesic distance on the relationship between structural covariance and genetic similarity. We found that the coupling between structural covariance and genetic similarity remained very strong after accounting for distance effects in this way. These results suggest that the relationship between the structural phenotypic covariance and genetic similarity is not simply explained by the physical distance between regional nodes in the networks. We have added these findings to the main manuscript and to supplemental information.

Changes to Results section of main paper

However, coupling between GS and SC remained strong even after controlling for the potentially confounding effects of geodesic distance by regression (for SA, $R(SC,GS) = 0.94$; for CT, $R(SC,GS) = 0.92$; and for NDI, $R(SC,GS) = 0.93$; all $P < 0.0001$) (Fig. S2).

Changes to Supplementary Information

3.1 Effects of physical distance on SC and GS

*We estimated Spearman's correlations between structural covariance and genetic similarity after accounting for linear effects of distance between each pair of structurally covarying or genetically similar regions nodes. As shown in **Figure S2**, the correlations between structural covariance and genetic similarity remained high, indicating that the strong coupling between phenotypic covariance and genetic correlation is not largely driven by the potentially confounding effect of physical distance between nodes.*

Figure S2. Distance effects on the relationship between structural covariance and genetic similarity. Edge-wise Spearman's correlation between genetic similarity (y-axis) and structural covariance (x-axis) matrices adjusted for distance effects.

2/5 The genetic results point most strongly to fetal brain development. How do the authors see this fitting with the typical age of onset for schizophrenia?

We thank the reviewer for their interesting question. Although schizophrenia has peak clinical incidence in late adolescence-early adult life, the psychotic disorder is generally regarded as neurodevelopmental in origin. Specifically, the neurodevelopmental model posits that genetic and early environmental factors (during pregnancy) perturb normal processes of brain development which predispose someone to develop schizophrenia symptoms later in life (Birnbaum & Weinberger, 2017). In line with this model, multiple epidemiological studies have reported increased risk of schizophrenia in adults who experienced environmental adversity (maternal infection, perinatal infection, complications of pregnancy) in late fetal life (Birnbaum & Weinberger, 2017; Khandaker, Zimbron, Lewis, & Jones, 2013; Rapoport, Giedd, & Gogtay, 2012; Weinberger, 1996). There are also data from animal models, e.g., maternal immune activation, indicating that schizophrenia-like phenotypes are more frequent in animals exposed to fetal or perinatal adversity (Canetta & Brown, 2012; F. L. Haddad, Patel, & Schmid, 2020). Additionally, polygenic scores for schizophrenia are associated with differences in early neurodevelopmental outcomes (Karcher et al., 2022; Riglin et al., 2017; Schlag et al., 2022). Collectively these data indicate that early brain development is critical for later risk of schizophrenia. This is compatible with our data showing (i) that genes significantly associated with schizophrenia or MRI metrics are highly expressed during the mid-gestation period (**Figure 1 C**) and (ii) that the genes which have the strongest pleiotropic association with schizophrenia and brain structure were enriched for processes relating to early neurodevelopment including neurogenesis and glial cell development. We have adapted the discussion as follows:

Changes to Discussion section of main text

These findings are in line with the neurodevelopmental model of schizophrenia, positing that genetic and early environmental factors perturb normal processes of brain development, including atypical formation of synaptic connections and axonal projections, with anatomically distributed effects on adult brain connectivity, that predispose individuals to develop psychotic symptoms later in life [8, 45, 46].

2/6 In the discussion it can be acknowledged that the analysis was limited to common SNPs, but rare genetic mutations also play a role in schizophrenia.

We agree with the reviewer that the analysis was limited to common genetic variants and that rare variants have been implicated in schizophrenia. We have updated the Discussion as follows:

These are plausible but preliminary results and further studies, also integrating rare variants, will be needed to survey the commonalities and differences between brain disorders in terms of their genetic relationships with brain structure.

References

- Birnbaum, R., & Weinberger, D. R. (2017). Genetic insights into the neurodevelopmental origins of schizophrenia. *Nature Reviews Neuroscience*, *18*(12), 727-740.
- Canetta, S. E., & Brown, A. S. (2012). Prenatal infection, maternal immune activation, and risk for schizophrenia. *Translational neuroscience*, *3*, 320-327.
- Grasby, K. L., Jahanshad, N., Painter, J. N., Colodro-Conde, L., Bralten, J., Hibar, D. P., . . . McMahon, M. A. B. (2020). The genetic architecture of the human cerebral cortex. *Science*, *367*(6484), eaay6690.
- Haddad, E., Pizzagalli, F., Zhu, A. H., Bhatt, R. R., Islam, T., Ba Gari, I., . . . Jahanshad, N. (2023). Multisite test-retest reliability and compatibility of brain metrics derived from FreeSurfer versions 7.1, 6.0, and 5.3. *Human Brain Mapping*, *44*(4), 1515-1532.
- Haddad, F. L., Patel, S. V., & Schmid, S. (2020). Maternal immune activation by poly I: C as a preclinical model for neurodevelopmental disorders: a focus on autism and schizophrenia. *Neuroscience & Biobehavioral Reviews*, *113*, 546-567.
- Hedges, E. P., Dimitrov, M., Zahid, U., Vega, B. B., Si, S., Dickson, H., . . . Kempton, M. J. (2022). Reliability of structural MRI measurements: The effects of scan session, head tilt, inter-scan interval, acquisition sequence, FreeSurfer version and processing stream. *Neuroimage*, *246*, 118751.
- Khandaker, G. M., Zimbron, J., Lewis, G., & Jones, P. (2013). Prenatal maternal infection, neurodevelopment and adult schizophrenia: a systematic review of population-based studies. *Psychological medicine*, *43*(2), 239-257.
- Knusmann, G. N., Anderson, J. S., Prigge, M. B., Dean III, D. C., Lange, N., Bigler, E. D., . . . King, J. B. (2022). Test-retest reliability of FreeSurfer-derived volume, area and cortical thickness from MPRAGE and MP2RAGE brain MRI images. *Neuroimage: Reports*, *2*(2), 100086.
- Rapoport, J., Giedd, J., & Gogtay, N. (2012). Neurodevelopmental model of schizophrenia: update 2012. *Molecular psychiatry*, *17*(12), 1228-1238.
- Trubetskov, V., Pardiñas, A. F., Qi, T., Panagiotaropoulou, G., Awasthi, S., Bigdeli, T. B., . . . Bertolino, A. (2022). Mapping genomic loci implicates genes and synaptic biology in schizophrenia. *Nature*. doi:10.1038/s41586-022-04434-5
- Warrier, V., Stauffer, E.-M., Huang, Q. Q., Wigdor, E. M., Slob, E. A., Seidlitz, J., . . . Grotzinger, A. D. (2022). The genetics of cortical organisation and development: a study of 2,347 neuroimaging phenotypes. *bioRxiv*, 2022.2009.2008.507084.
- Weinberger, D. R. (1996). On the plausibility of “the neurodevelopmental hypothesis” of schizophrenia. *Neuropsychopharmacology*, *14*(1), 1-11.
- Werme, J., van der Sluis, S., Posthuma, D., & De Leeuw, C. (2021). LAVA: An integrated framework for local genetic correlation analysis. *bioRxiv*, 2020.2012.2031.424652.
- Karcher, N. R., Paul, S. E., Johnson, E. C., Hatoum, A. S., Baranger, D. A., Agrawal, A., . . . Bogdan, R. (2022). Psychotic-like experiences and polygenic liability in the adolescent brain cognitive development study. *Biological Psychiatry: Cognitive Neuroscience and Neuroimaging*, *7*(1), 45-55.
- Riglin, L., Collishaw, S., Richards, A., Thapar, A. K., Maughan, B., O'Donovan, M. C., & Thapar, A. (2017). Schizophrenia risk alleles and neurodevelopmental outcomes in childhood: a population-based cohort study. *The Lancet Psychiatry*, *4*(1), 57-62.
- Schlag, F., Allegrini, A. G., Buitelaar, J., Verhoef, E., van Donkelaar, M., Plomin, R., . . . St Pourcain, B. (2022). Polygenic risk for mental disorder reveals distinct association

profiles across social behaviour in the general population. *Molecular psychiatry*, 27(3), 1588-1598.

Reviewer #1 (Remarks to the Author):

Stauffer et al have done extensive analyses and revision and addressed the comments adequately, with a couple of exceptions:

1. Selecting schizophrenia genes. They argue that there are several methods, and it is of interest that they argue against the methods used in the original Trubetskoy publication which the current analyses build on. However, the main problem with the current analyses remains: the phenotype of interest, schizophrenia, is highly polygenic, and nearly half of all human genes are expressed in the brain. Thus, the current approach will by chance generate lists of brain-associated genes, and it is likely that approx. 40%-46% of genes from Trubetskoy will overlap with their or any other selections of schizophrenia genes. This has to be included as a limitation.
2. They find robust evidence for chromosome 17q21, which should be emphasized more as a robust finding. Psychiatric genetics have a long history of non-replication (PMID: 25754081), and it is important to avoid repeating the mistakes from the past when combining with imaging-genetics analyses.
3. The results from the investigation of the specificity of the results are interesting (Bipolar, Alzheimer). However, they did not include the requested phenotype height. It is of importance to compare with a non-brain, non-mental disorder phenotype, and they overlooked this request in their response. It is critical to compare their gene overlap findings in schizophrenia with height.

Reviewer #2 (Remarks to the Author):

Many thanks to the authors for addressing the points I raised previously.

I think one issue remains. I suggested to check whether repeatability of measurement across brain regions might have affected which regions are identified as 'hubs'. The authors responded that the data were carefully controlled to exclude noise, and that test-retest reliability of measurement has been shown to be good-to-excellent in these data and for these specific measures. While this is reassuring, it still leaves open the possibility that measurement reliability relates to hubness. 'Good-to-excellent' implies a range of different degrees of reliability across regions, so the question is whether there is a quantitative relationship between hubness and reliability within this range. From what the authors wrote, it should be easy to assess this given the available data. It seems an important issue because the hubs are often taken as important biologically, so the reader would be reassured to know that they are not reflections of measurement reliability.

Point-by-point response

We thank the two reviewers for their positive feedback and for their suggestions. In this revision, we have carefully considered each point raised and have incorporated relevant amendments into the manuscript. Changes and additional analyses to the revised manuscript are highlighted in the response.

Reviewer #1 (Remarks to the Author):

Stauffer et al have done extensive analyses and revision and addressed the comments adequately, with a couple of exceptions:

1/1 Selecting schizophrenia genes. They argue that there are several methods, and it is of interest that they argue against the methods used in the original Trubetskoy publication which the current analyses build on. However, the main problem with the current analyses remains: the phenotype of interest, schizophrenia, is highly polygenic, and nearly half of all human genes are expressed in the brain. Thus, the current approach will by chance generate lists of brain-associated genes, and it is likely that approx. 40%-46% of genes from Trubetskoy will overlap with their or any other selections of schizophrenia genes. This has to be included as a limitation.

We thank the reviewer for this comment, and we agree that schizophrenia is highly polygenic and that a large number of genes is expressed in the brain. We would like to clarify that it was not our intention to argue *against* the methods used in the original Trubetskoy publication. On the contrary, we think that the diverse methods used by Trubetskoy are of the highest quality, which is why we additionally tested whether the genetic effects of surface area, cortical thickness and neurite density index were enriched for high-confidence schizophrenia-related genes identified by Trubetskoy et al. As outlined in our first response, we found significant enrichments for surface area and cortical thickness. However, as suggested by the reviewer, we included this as a limitation in our manuscript.

Changes to Discussion

These are plausible but preliminary results and several limitations need to be considered. First, this study was limited to common variants. However, it is known that schizophrenia is additionally associated with rare variants [9]. Second, the PLS analyses were based on a specific SNP-to-gene mapping method (i.e. H-MAGMA). Further studies, also integrating rare variants, and multiple SNP-to-gene mapping methods, will be needed to survey the commonalities and differences between brain disorders in terms of their genetic relationships with brain structure.

1/2 They find robust evidence for chromosome 17q21, which should be emphasized more as a robust finding. Psychiatric genetics have a long history of non-replication (PMID: 25754081), and it is important to avoid repeating the mistakes from the past when combining with imaging-genetics analyses.

We thank the reviewer for this suggestion. We have now emphasized that genes within chromosome 17q21 were robustly identified using different approaches.

Changes to the Discussion

...Genes within 17q21 were consistently identified across the different methodological approaches. Encouragingly, genetic variation in the 17q21 region has been replicably associated with various measures of brain structure [56 , 57 , 58, 59], as well as with schizophrenia [27, 60], in prior studies. However, chromosome 17q21 has also been associated with other disorders, such as autism spectrum disorder [61] and Alzheimer's disease [62], suggesting that this region might have effects on brain phenotypes that contribute to the pathogenesis of several neuropsychiatric diseases.

1/3 The results from the investigation of the specificity of the results are interesting (Bipolar, Alzheimer). However, they did not include the requested phenotype height. It is of importance to compare with a non-brain, non-mental disorder phenotype, and they overlooked this request in their response. It is critical to compare their gene overlap findings in schizophrenia with height.

We thank the reviewer for their comment, and we are pleased that the specificity results using bipolar disorder and Alzheimer's disease generated interest. As requested by the reviewer, we repeated the principal analyses using height as a phenotype. Height is a neurodevelopmentally sensitive phenotype and has been shown to be phenotypically and genetically correlated with various MRI metrics, including surface area (Grasby et al., 2020; Hofer et al., 2020; Tilot et al., 2021; Vuoksima et al., 2018). In line with previous findings of genetic correlations between height and MRI phenotypes, we found that height showed significant covariance with all MRI metrics. MRI covariation explained approximately the same proportion of variance in height as it did for schizophrenia risk. However, the pleiotropic genes identified for MRI metrics and height were largely distinct from pleiotropic genes associated with MRI metrics and schizophrenia (less than 50% overlap). Thus, while there was significant genetic covariation between MRI metrics and both schizophrenia and height, these were driven by different genes. We have now integrated these findings in the main manuscript and supplemental materials, as excerpted verbatim below.

Changes to the Results

Clinical diagnostic specificity of genetic covariation between schizophrenia and brain structure

We repeated many of the principal analyses of pleiotropic association with schizophrenia, using identical methods and models applied to independent large-scale GWAS data for two additional neuropsychiatric disorders - bipolar disorder (BIP) (N = 41,917 cases and N = 371,549 controls) [41] and Alzheimer's disease (AD) (N = 398,058) [42] - and for height, a neurodevelopmentally sensitive non-psychiatric phenotype (N = 4,080,687) [43]. We identified genes significantly associated with BIP, AD or height and investigated their intersection with MRI-associated genes. Out of 136 BIP-associated genes, only the intersection with 15 genes also associated with SA was significant, and largely comprised genes located at chromosome

3p21. Out of 77 AD-associated genes, there were significant intersections with genes also associated with SA (15), CT (12), and NDI (11); and several genes associated with both AD and surface area were located at chromosome 17q21. Out of 8012 genes associated with height, 175 were associated with SA, 55 with CT and 23 with NDI; 21 genes were shared between height and all 3 MRI metrics, and these were located on chromosome 17q21, 8p23 and chromosome 1p33. We also used PLS regression, as previously for analysis of whole-genome covariation with schizophrenia (Fig. 2), for comparable analysis of regional brain phenotypes that were pleiotropically associated with risk of BIP, AD or height; see Fig.6 and SI Results 6 for details.

Figure 6. Specificity of pleiotropic associations between clinical disorders or height and regional brain phenotypes. (A) Proportion of variance in the genetically predicted risk for each disorder and height (y-axis) explained by the genetic effects on regional MRI metrics (x-axis; SA = surface area, CT = cortical thickness, NDI = neurite density index) based on the first PLS component, PLS1. The proportion of disorder-related variance explained was greater for schizophrenia (SCZ; SA = 5.9%, CT = 5.5%, NDI = 3%), than for bipolar disorder (BIP; SA = 2.7%, CT = 3.6%, NDI = 1.6%), or Alzheimer's disease (AD; SA = 1.5%, CT = 1.2%, NDI = 0.8%). The proportion of height-related variance was comparable to the proportion of schizophrenia-related variance across all MRI metrics. (B) Cortical surface maps of PLS1 regional brain weights SCZ, BIP, AD and height. Higher positive weights (shades of yellow) indicate stronger genetic covariation with each disorder; regions with zero weight are shown in white. Mean absolute weights were lower for BIP (SA \bar{w} = 12.3, CT \bar{w} = 9.45, NDI \bar{w} = 8), and for AD (SA \bar{w} = 9, CT \bar{w} = 5.2, NDI \bar{w} = 5.4), than for schizophrenia (SA \bar{w} = 18.29, CT \bar{w} = 11.89, NDI \bar{w} = 11.37). Apart from SA, mean PLS weights for height were generally lower than for schizophrenia (SA \bar{w} = 20.4, CT \bar{w} = 12.1, NDI \bar{w} = 10.5). Fewer brain regions had significant PLS1 scores for BIP (NDI = 175) and AD (SA = 79, CT = 166, NDI = 170) than for schizophrenia (SA, CT = 180, NDI = 179). For height, all brain regions showed significant PLS1 scores. (C) Spearman's correlations (ρ ; y-axis) between T and U scores for schizophrenia, bipolar disorder, Alzheimer's disease and height. The strength of pleiotropic association indexed by ρ was greater for schizophrenia (SA ρ = 0.24, CT ρ = 0.23, NDI ρ = 0.17), than for BIP (SA ρ = 0.17, CT

$\rho = 0.19$, NDI $\rho = 0.13$), AD (SA $\rho = 0.12$, CT $\rho = 0.11$, NDI $\rho = 0.09$). For SA, the pleiotropic association with height was stronger compared to schizophrenia (SA $\rho = 0.27$, CT $\rho = 0.23$, NDI $\rho = 0.16$). (D) Venn diagrams showing the intersection of the top 1% most pleiotropic genes, with the highest $\Delta(R(T,U))$ scores, for each MRI metric. The proportion of overlapping genes was less than 50% for bipolar disorder and schizophrenia and further decreased for either Alzheimer's disease or height and schizophrenia. This implied that the genes identified for MRI metrics and schizophrenia are largely distinct from the genes identified for Alzheimer's disease or height.

Changes to Supplementary Information, Results

6 Specificity of schizophrenia's intersection with MRI-associated genes

...For height, we identified 8,012 genes after accounting for multiple comparison correction. The height GWAS was based on subjects of predominantly European ancestries and represents one of the largest, most well-powered GWAS's to date [17]. Out of 8,012 genes associated with height, 221 were also associated with SA, 96 genes with CT and 54 genes with NDI, representing a significant overlap. 21 genes were shared between height, SA, CT and NDI and were found on chromosome 17q21, 8p23 and chromosome 1p33.

As outlined in the main manuscript, the first PLS component for each MRI metric identified a small but significant proportion of its genetically determined variation that covaried with genetic risks for bipolar disorder (2.7% for SA, 3.6% for CT and 1.6% for NDI). However, the variance explained for BIP was about 50% less than the variance explained for schizophrenia by the same MRI metrics; and the variance explained for AD was about 75% less than for schizophrenia (1.5% for SA, 1.2% for CT and 0.8% for NDI). The proportion of height-related variance was comparable to the proportion of schizophrenia-related variance across all MRI metrics (Height; SA = 7.1%, CT = 5.5%, NDI = 2.7%) (Fig.6 A).

The strength of pleiotropic association with schizophrenia and MRI phenotypes, across all 18,640 genes, was also greater for schizophrenia than for BIP or AD. Specifically, the correlation between T and U scores decreased from schizophrenia to bipolar disorder, to Alzheimer's disease. For height, we again found that the strength of pleiotropic association with SA was higher compared to schizophrenia, but similar to or lower than the strength of pleiotropic association with schizophrenia for CT and NDI (Figure 6 C). Finally, we visualised the intersection of the top 1% of genes with the highest $\Delta(R(T,U))$ values between schizophrenia and bipolar disorder, Alzheimer's disease, or height. The proportion of overlapping genes was less than 50% for bipolar and schizophrenia, and further decreased for Alzheimer's disease and schizophrenia, or for height and schizophrenia (Fig. 6 D). In summary, these results suggest that the genetic covariation between brain structure and schizophrenia is stronger than the genetic covariation between brain structure and bipolar disorder or Alzheimer's disease for all MRI metrics; and stronger than the genetic covariation between brain structure and height for most MRI metrics. More than 50% of the genes investigated in down-stream analysis were specific to schizophrenia. In this context we note that the top 1% of genes with the highest $\Delta(R(T,U))$ for height and CT, SA and NDI were not enriched for constrained genes ($P > 0.05$) compared to schizophrenia (Fig. S9).

Based on the results on the genetic relationship between height and SA, CT or NDI, we were stimulated to investigate the phenotypic relationship between these MRI metrics and height in our sample. To this end, we correlated the global MRI metrics of SA, NDI and CT with standing height measured in cm. In line with previous studies, we found a strong positive correlation between height and SA, and no significant correlation between height and CT or NDI [18, 19, 20, 21].

Figure S10. Phenotypic relationships between height and MRI metrics. Shown are Spearman's correlations between standing height (y-axis, cm) and global surface area (SA, mm²), cortical thickness (CT, mm) and neurite density index (NDI, density in %). Each point represents one of 31,780 subjects included in the main analyses.

Changes to Discussion:

...Genes within 17q21 were consistently identified across the different methodological approaches. Encouragingly, genetic variation in the 17q21 region has been replicably associated with various measures of brain structure [56, 57, 58, 59], as well as with schizophrenia [27, 60], in prior studies. However, chromosome 17q21 has also been associated with other disorders, such as autism spectrum disorder [61] and Alzheimer's disease [62], suggesting that this region might have effects on brain phenotypes that contribute to the pathogenesis of several neuropsychiatric diseases.

...

...For SA we found that its genetic covariation was somewhat greater with height compared to its genetic covariation with schizophrenia. This finding is in line with studies reporting phenotypic and genetic correlations between height and surface area [1, 76, 77, 78], as well as between height and schizophrenia [79, 80]. It is also consistent with our finding that height was strongly positively correlated with SA (but not CT or NDI) in these data (Fig S10). This implies that genetic variants located at 17q21 may have normative effects on the correlated phenotypes of height and brain surface area as well as conferring increased risk of schizophrenia.

Reviewer #2 (Remarks to the Author):

Many thanks to the authors for addressing the points I raised previously. I think one issue remains.

2/1 I suggested to check whether repeatability of measurement across brain regions might have affected which regions are identified as 'hubs'. The authors responded that the data were carefully controlled to exclude noise, and that test-retest reliability of measurement has been shown to be good-to-excellent in these data and for these specific measures. While this is reassuring, it still leaves open the possibility that measurement reliability relates to hubness. 'Good-to-excellent' implies a range of different degrees of reliability across regions, so the question is whether there is a quantitative relationship between hubness and reliability within this range. From what the authors wrote, it should be easy to assess this given the available data. It seems an important issue because the hubs are often taken as important biologically, so the reader would be reassured to know that they are not reflections of measurement reliability.

We thank the reviewer for their suggestion to investigate the relationship between test-retest reliability and regional degree (hubness) to ensure that regional degree is not simply a reflection of measurement reliability. To make this analysis directly relevant to our results, we focused on the two macrostructural phenotypes, surface area and cortical thickness, for which test-retest data were available from the UK Biobank cohort. We did not find a significant correlation between measurement reliability of these metrics and regional degree of the structural covariance networks derived from them. We integrated these findings in the main manuscript and the supplement as outlined below.

Changes to the Supplement

3.4 Effects of measurement reliability on regional degree

To ensure that regional degree (hubness) was not merely a reflection of measurement reliability, we investigated the relationship between measurement reliability and regional degree in a subset of the UK Biobank imaging sample. To this end, we assessed a subset of subjects from the UK Biobank for whom repeat scans were available (N = 1,363). Only T1 follow-up scans were only available for this analysis, thus we estimated CT and SA. We performed the same imaging quality controls as for the baseline scans. We excluded subjects with incomplete imaging data and subjects that were identified as global or regional outliers, i.e. more than five times the median absolute deviation from the sample median. For CT and SA, this led to N = 1,280 subjects with both baseline and follow-up scans. To assess test-retest reliability of the regional MRI metrics, we estimated the (Pearson's) correlation between the group average baseline and follow-up measurements of SA or CT for each brain region. For CT, the test-retest reliability ranged between $0.5 \geq R \leq 0.9$ (mean $R = 0.78$) and for SA $0.71 \geq R \leq 0.99$ (mean $R = 0.97$). Thus, in line with previous findings on these metrics, the test-retest reliability was good-to-excellent (Duff et al., 2022). Finally, we calculated Pearson's correlations between test-retest reliability and regional degree based on structural covariance networks derived from the baseline CT or SA data. As shown in Figure S5, there was no significant correlation between measurement reliability and regional degree (CT, $R = 0.1$, $P =$

0.17; SA, $R = -0.06$, $P = 0.4$). These results suggest that regional degree is not driven by test-retest reliability of MRI metrics.

Figure 5. Measurement reliability and regional degree. Scatterplots of measurement reliability, indexed by Pearson’s correlations between baseline and follow-up scans (y-axis), and regional degree based on structural covariance networks (x-axis) for cortical thickness (CT) and surface area (SA). Each data point represents one of 180 brain regions.

Changes to the Methods

To ensure that regional degree was not driven by measurement reliability, we used follow-up scans from a subset of the UK Biobank cohort ($N = 1,280$) to investigate the relationship between test-retest reliability of SA and CT and weighted degree of regional nodes in the corresponding structural covariance networks. We found no significant correlation between measurement reliability and hubness indexed by regional degree for either of these metrics (see SI Results 3.4 for details).

References

- Duff, E., Zelaya, F., Almagro, F. A., Miller, K. L., Martin, N., Nichols, T. E., . . . Douaud, G. (2022). Reliability of multi-site UK Biobank MRI brain phenotypes for the assessment of neuropsychiatric complications of SARS-CoV-2 infection: The COVID-CNS travelling heads study. *Plos one*, 17(9), e0273704.
- Grasby, K. L., Jahanshad, N., Painter, J. N., Colodro-Conde, L., Bralten, J., Hibar, D. P., . . . McMahon, M. A. B. (2020). The genetic architecture of the human cerebral cortex. *Science*, 367(6484), eaay6690.
- Hofer, E., Roshchupkin, G. V., Adams, H. H., Knol, M. J., Lin, H., Li, S., . . . Satizabal, C. L. (2020). Genetic correlations and genome-wide associations of cortical structure in general population samples of 22,824 adults. *Nature communications*, 11(1), 4796.
- Tilot, A. K., Khramtsova, E. A., Liang, D., Grasby, K. L., Jahanshad, N., Painter, J., . . . Lind, P. A. (2021). The evolutionary history of common genetic variants influencing human cortical surface area. *Cerebral Cortex*, 31(4), 1873-1887.
- Vuoksima, E., Panizzon, M. S., Franz, C. E., Fennema-Notestine, C., Hagler, D. J., Lyons, M. J., . . . Kremen, W. S. (2018). Brain structure mediates the association between height and cognitive ability. *Brain Structure and Function*, 223, 3487-3494.

REVIEWERS' COMMENTS

Reviewer #1 (Remarks to the Author):

The authors have adequately addressed the comments

Reviewer #2 (Remarks to the Author):

Thank you for addressing my final query. No further comments.